# How Far Have We Gone in Vulnerability Detection Using Large Language Models

## Abstract

As software becomes increasingly complex and prone to vulnerabilities, automated vulnerability detection is critically important, yet challenging. Given the significant successes of Large Language Models (LLMs) in various tasks, there is growing anticipation of their efficacy in vulnerability detection. However, a quantitative understanding of their potential in vulnerability detection is still missing. To bridge this gap, we introduce a comprehensive vulnerability benchmark `VulBench`. This benchmark aggregates high-quality data from a wide range of CTF (Capture-the-Flag) [1] challenges and real-world applications, with annotations for each vulnerable function detailing the vulnerability type and its root cause. Through our experiments encompassing 16 LLMs and 6 state-of-the-art (SOTA) deep learning-based models and static analyzers, we find that several LLMs outperform traditional deep learning approaches in vulnerability detection, revealing an untapped potential in LLMs. This work contributes to the understanding and utilization of LLMs for enhanced software security.

## 1 Introduction

The rapid advancement of software systems has led to an increase in their complexity and susceptibility to vulnerabilities, thereby heightening security risks. Effective vulnerability detection is imperative in this scenario, necessitating robust and automated methods. Traditional techniques like fuzzing, symbolic execution, and static analysis, while valuable, often fall short in addressing the complexities of modern software systems (AFL; Cadar et al., 2008).

In this context, Large Language Models (LLMs) such as GPT-3.5 and GPT-4 have emerged as promising tools. Noted for their broad generalization and reasoning capabilities, these models have shown notable success in diverse applications, including the domain of vulnerability detection (OpenAI; Yang et al., 2023b). Yet, the quantitative assessment of their effectiveness in vulnerability detection is still underexplored.

A significant obstacle in applying LLMs for vulnerability detection is the lack of high-quality, accurate datasets. Prior research shows that existing vulnerability datasets often lack in quality and accuracy, achieving detection correctness rates as low as 30% to 50% (Croft et al., 2023; Chen et al., 2023). This not only hampers the ability to effectively evaluate LLMs but also fails to represent the complexities of real-world software vulnerabilities, which often arise from interactions across multiple functions and require a comprehensive understanding of the codebase.

To address these challenges, we introduce `VulBench`, a comprehensive dataset amalgamating data from various sources, including CTF challenges, MAGMA (Hazimeh et al., 2020), Devign (Zhou et al., 2019), D2A (Zheng et al., 2021), and Big-Vul (Fan et al., 2020). This dataset offers a blend of straightforward CTF challenges and more complex real-world CVE vulnerabilities, catering to both open-source and closed-source software vulnerability detection scenarios. All datasets are expertly human-labeled, with the CTF and MAGMA datasets additionally providing the necessary context for vulnerability assessment.

We have designed and conducted a comprehensive evaluation process to assess the vulnerability detection capabilities of LLMs. Our assessments are grounded on the multi-level vulnerability data

---

[1] https://ctf-wiki.org/en/

provided within these datasets. Employing two approaches—binary classification to determine the presence of vulnerabilities within functions, and multi-class classification to identify specific types of function vulnerabilities—we facilitate nuanced and in-depth judgments by the models. This evaluative methodology is consistently applied to both deep learning models and static analysis tools, ensuring a uniform standard of evaluation across different systems and models.

Our main contributions are as follows:

- We conduct the first large-scale study to quantitatively measure the performance of 16 LLMs in the field of vulnerability detection, setting a benchmark against state-of-the-art deep learning models and static analyzers.
- The introduction of `VulBench`, a benchmark that addresses the quality issues prevalent in existing datasets, offering a comprehensive dataset for more accurate evaluations, along with the new natural language description for the vulnerabilities.
- Unveiling the untapped potential of LLMs in vulnerability detection, our findings provide new insights and future research directions in this domain.
- We provide the dataset in Anonymous Github to facilitate future work.

This research not only enhances our understanding of LLMs' application in software security but also opens up new avenues for advancements in automated vulnerability detection.

## 2 RELATED WORK

### 2.1 BACKGROUND FOR VULNERABILITY DETECTION

Vulnerability detection is a crucial task in the field of computer security. Its primary objective is to identify potential software security threats, thus reducing the risk of cyber-attacks. A key resource in this effort is the CVE database (MITRE), which acts as a platform for monitoring these vulnerabilities.

```
1   int main() {
2       char buf[10];
3       char str[10];
4       scanf("%5s", str);
5       gets(buf)
6       if(str[9] == 'a')
7           system("/bin/sh");
8       else
9           puts("Finished!");
10      return 0;
11  }
```

Listing 1: Vulnerability example with a 'Stack Overflow' Vulnerability.

Three principal techniques are employed in vulnerability discovery: fuzzing, symbolic execution, and static analysis. Fuzzing (AFL) seeks to uncover software crashes and anomalies by inundating the system with diverse random inputs. Conversely, symbolic execution (Cadar et al., 2008) aims to detect irregularities by simulating multiple application pathways. Finally, static analysis (Lu et al., 2019; Wu et al., 2021) examines the code without executing it to identify potential vulnerabilities.

Integrating deep learning models into vulnerability exploration often involves feeding source code into the model for classification (Lu et al., 2021; Hanif & Maffeis, 2022). This approach, which analyzes the code without execution, is generally classified under static analysis. Nevertheless, significant advancements have also been made in utilizing deep learning models to augment fuzzing techniques (Shi et al., 2023; Godefroid et al., 2017).

Listing 1 is an example of a vulnerable function. The data in `char buf[10]` will overflow into `str[10]` due to the unsafe usage of `gets(buf);`. The duty of vulnerability detection is to detect that potential stack overflow.

### 2.2 DEEP LEARNING-BASED MODELS FOR VULNERABILITY DETECTION

This research builds upon recent strides in NLP-driven code analysis for vulnerability detection. CodeXGLUE (Lu et al., 2021) works with CodeBERT on the Devign (Zhou et al., 2019) dataset represents a key development in assessing source code vulnerability risk. LineVul (Fu & Tantithamthavorn, 2022) extends these insights by applying the same model to the Big-Vul dataset for nuanced

detection at both function and line levels. Alternatively, VulBERTa (Hanif & Maffeis, 2022) innovates with a RoBERTa (Liu et al., 2019) model tailored for code through a hybrid BPE-tokenization scheme. The VulDeePecker (Li et al., 2018; Zou et al., 2019; Li et al., 2021) series introduces and iteratively refines the idea of utilizing semantically correlated "code gadgets" and BLSTMs for initial vulnerability identification, then multi-class categorization, and finally precise location via LLVM IR analysis. ReVeal (Chakraborty et al., 2022) rounds out the landscape by proving that the efficacy of vulnerability prediction can be enhanced through the integration of semantic information using gated graph neural networks (GGNN), combined with refined data handling practices.

### 2.3 LARGE LANGUAGE MODELS FOR VULNERABILITY DETECTION

Previous research (OpenAI) has demonstrated the potential of LLMs for detecting software vulnerabilities, although there is a lack of comprehensive vulnerability-focused data. While GPT-3.5 and GPT-4 have been the primary subjects of study, displaying notable capabilities, evaluations of open-access LLMs are less common. Studies present mixed results (Cheshkov et al., 2023); while some LLMs performed on par with naive classifiers, others, specifically GPT-4, have shown significant advantages over conventional static analysis tools (Noever, 2023). Enhancements to exploit the multi-round dialogue proficiency of ChatGPT have prompted improved detection methods (Zhang et al., 2023), and recent advancements (Chan et al., 2023) have fine-tuned LLMs for identifying diverse vulnerability patterns, indicating a potential for reaching expert-level performance in vulnerability management tasks.

### 2.4 BENCHMARKS FOR LLMS

Open LLM Leaderboard (Beeching et al., 2023) and MMLU (Hendrycks et al., 2021) evaluates models on science questions, commonsense inference, etc. MT-bench (Zheng et al., 2023) evaluates chat assistants in aspects of writing, reasoning, code, and so on. Cheshkov et al. (2023) conducts the evaluation of vulnerability detection capability on the OpenAI GPT series. Besides, there are several individual datasets (Zheng et al., 2021; Zhou et al., 2019; Chen et al., 2023) for vulnerability detection.

## 3 DATASET

Contrary to past works that assemble vast datasets with multiple vulnerability types automatically, our focus hinges on the enhancement of the accuracy and validity of datasets. We've gathered from multiple sources relatively condensed yet comprehensive CTF datasets detailing all functions in an executable binary, and real-world datasets providing only partial functions from huge real-world programs. Although CTF problems don't originate from the real world, their associated vulnerabilities typically mirror real-world scenarios, presenting a miniature depiction of the real world.

### 3.1 DATASET OVERVIEW

The dataset amalgamates multiple sources of vulnerabilities, comprising CTF challenges, MAGMA (Hazimeh et al., 2020), and three previous vulnerability datasets (Zheng et al., 2021; Zhou et al., 2019; Fan et al., 2020) with extensively cleaning. They can be mainly divided into three types of data sets as follows. The overview of the dataset is shown in Table 1 and the containing vulnerability types are shown in Table 6.

**CTF Dataset**: In CTF challenges, particularly the PWN category, participants must identify and exploit program vulnerabilities to get the Flag, which serves as evidence of task completion. Despite being shorter than vulnerabilities found in real-world CVEs, there's a significant number of flawed functions within CTF PWN tasks, encompassing a broad range of potential memory-related issues flagged within CVEs. This makes it a suitable and basic method for evaluating the LLM.

**CVE Dataset**: The CVE dataset typically identifies vulnerable functions by comparing changes made to the code during CVE remediation. This information, derived from real-world software, includes not just memory leak vulnerabilities typical to CTF scenarios but a broader range of real-world vulnerabilities, providing a more strenuous test of LLM's vulnerability detection capabilities.

| Type | Obtaining Method | Source | Features | Count | #Label |
|------|------------------|--------|----------|-------|--------|
| CTF | PWN Problem | BUUOJ | Raw decompiled code, Reversed decompiled code | 108 | 5 |
| Real World | CVE Commit Diff | MAGMA | Source code, Raw decompiled code | 100 | 8 |
| | | Devign | Source code | 70 | 7 |
| | | Big-Vul | | 108 | 9 |
| | Static Analyzer | D2A | Source code | 69 | 4 |

Table 1: Composition and source of the dataset, along with the features available for the model and the count of binaries or functions in the datasets. In the CTF dataset, the count represents the count of individual binary, and there exist multiple functions in a binary. In real-world datasets, the count represents the count of individual bugs, and there exist multiple functions in MAGMA. #Label refers to the number of different types of vulnerabilities used for multi-label classification in each dataset.

**Static Analyser Generated Dataset**: The dataset of potential vulnerabilities, garnered via static analysis, is typically produced using specific tools for this analysis. It involves examining the entirety of a project, where possible weak points are detected by the patterns and constraints. Despite its usefulness, this method usually generates a relatively high rate of false positives.

## 3.2 DATASET CONSTRUCTION

### 3.2.1 CTF

We create a dataset for CTF challenges within the PWN category, by selecting problems from the BUUOJ platform [2]. In the CTF, participants are usually only provided with a binary, devoid of source code. To cater to Language Models such as LLM, which are less adept at handling assembly code directly, we use IDA (Interactive Disassembler) (Hex-Rays) to extract more interpretable decompiled code for model input. This is further complicated by the characteristics of the binaries in CTF competitions. They often lack essential debug and symbolization information, leaving them devoid of meaningful structure and variable definitions.

To mitigate this, we engage in manual reverse engineering to enhance the structure of the decompiled code by restoring recognizable constructs, renaming variables, and annotating the size of global variables. It's worth noting that, despite acquiring refined decompiled code via manual reverse engineering, the generated decompiled code often bears intrinsic patterns or code snippets characteristic from the decompiler, which are rare in the LLM's training set. As such, our efforts should focus on optimizing the readability of the decompiled code for LLM and evaluate the utility of such manual interventions in vulnerability detection, instead of aiming to enhance the decompiled code to mirror the actual source code distribution. We give an example in Section E.1 to demonstrate the difference between raw decompiled code and manually reversed decompiled code and the challenges of using decompiled code directly.

As for the metric, aside from the traditional binary classification and multi-class classification, we describe the root causes of identified vulnerabilities using clear, descriptive natural language explanations over the ambiguous practice of referencing code lines. An example is shown in Table 7.

### 3.2.2 MAGMA

Our study also adopts the MAGMA fuzzing dataset, a specialized collection created to assess the ability of fuzzing tools to trigger hidden vulnerabilities. In the fuzzing process, fuzzers create myriad testcases by mutating inputs randomly and feeding them into the target program. As vulnerabilities do not consistently lead to program crashes, relying solely on the execution state of the program is not an ideal method for detecting whether the vulnerability has been triggered. To address this, the MAGMA dataset includes specialized markers—referred to as 'canaries'—on the execution paths towards the location of vulnerability. When a canary's check returns true, it implies that the vulnerability has been triggered. This approach does not depend on the more rigorous occurrence of

---
[2]https://buuoj.cn

program crashes to ascertain the presence of vulnerabilities. When scrutinizing vulnerabilities, we analyze the security patches and 'canaries' to understand the root cause of the flaws. In Section E.2, we give an example of how we utilize this information.

To enhance the analysis, we provide additional context such as macro expressions and correlatable functions related to the vulnerability during the analysis. This additional context is used to simulate a real-world scenario where understanding the relationships between functions can prove pivotal in identifying security vulnerability so that we can assess the model's capability when more context is provided and its resistance to the impact of extraneous information. Also, we provide two input features for LLM, the source code extracted from project repositories and the decompiled code extracted from the compiled binary to mimic the scenarios of closed-source software.

### 3.2.3 DEVIGN, D2A, BIG-VUL

For the purpose of supplementing our research with real-world vulnerability data, we additionally incorporate three prior datasets. These include the D2A dataset proposed by Zheng et al. (2021), the Devign dataset introduced by Zhou et al. (2019), and the Big-Vul dataset discussed in Fan et al. (2020). Nevertheless, we don't directly employ these datasets in their entirety. Rather, we take guidance from previous studies to ensure maximum accuracy and reliability within our dataset.

In Croft et al. (2023), a selection of vulnerability functions are randomly sampled from the triad of datasets. Those are then manually evaluated to ascertain if they represented authentic security patches within the git commit to mitigate the interfering factors such as non-functional changes in the same commit and wrongly identified due to unreliable keyword matching or false positives from static analysis tools. When constructing the dataset, functions verified as true security patches (Related to fixing the vulnerability) are marked as vulnerable. On the other hand, functions that don't qualify as security patches (Related to code cleanup or irrelevant code changes) are designated as non-vulnerable. Furthermore, we also consider patched functions to be invulnerable.

In the annotation process, for Big-Vul dataset, we fuse the descriptions from their CVE pages with existing tags to categorize the types and we illustrate the information provided by Big-Vul dataset and the annotation process in Section E.3. For Devign dataset, it lacks corresponding vulnerability types, leading to our reliance on CVE descriptions to distinguish the vulnerability types. As for the D2A vulnerability dataset, which obtained using a static analysis program that automatically notes the vulnerability type, it has already been tagged by static analysis tools, facilitating our direct reuse.

## 4 EVALUATION

### 4.1 SELECTED MODELS & BASELINES

In our experiment, we select a series of large models, including GPT-3.5, GPT-4 and open-access LLMs comprises variations of Llama2 and those that underwent SFT on Llama2. The full list of selected models is in Table 5. We host the models on 48 A800 GPUs across 6 nodes and leverage vLLM (Kwon et al., 2023) and text-generation-inference (Huggingface) to expedite the model inference process. We repeat the requests for 5 times for each vulnerability detection task.

For comparison, we select three deep learning models and three rule-based static analysis tools according to Steenhoek et al. (2023); Lipp et al. (2022) to serve as baselines. The full list and description are shown in the Section B. We exclude the VulDeeLocator (Li et al., 2021; Zou et al., 2019) as it requires a lot of human effort to annotate the execution trace and requires the source code compile-able. Using these, we aim to ascertain the degree of superiority, in terms of vulnerability detection, the current LLM possesses compared to traditional methods. Especially, the selected deep learning models can only perform binary classification as it requires retraining to support multi-class classification, but the size of our dataset is too limited to support it.

### 4.2 METRIC

Unlike GPT models that boast of an excellent alignment (Ouyang et al., 2022), other models don't come equipped with a mechanism to seamlessly output in a standard format. To address this situation, we employ the two few-shot methods, 2-shots and 5-shots, to insist on a uniform template for

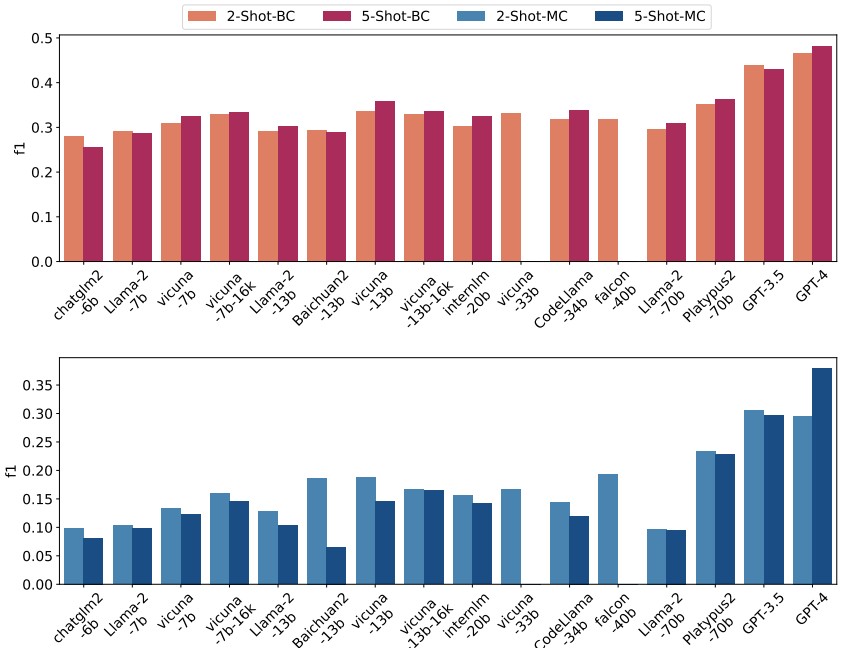

Figure 1: Few-shot conversation for binary classification (whether a function is vulnerable) and multi-class classification (which vulnerability does the function have) in CTF dataset. `BC` stands for binary classification. `MC` stands for multi-class classification.

output, making it easier to parse the answer of models. Within all the datasets, we randomly select the examples that serve as our few-shot instances. The prompts are formatted in the style of a chat to ensure alignment with the models' methods. Each dialogue round incorporated a few-shot instance, with the substantive question introduced only from the third round onwards. Figures 3 and 4 offer a clear showcase of the 2-shot prompt at work.

During binary classification, the model is programmed to channel outputs as `VULNERABLE: NO` or `VULNERABLE: YES`. Alternately, in multi-class classification scenarios, the model delivers outputs as `TYPE: [type]`, where `[type]` corresponds to a selection from an exhaustive list. To minimize parsing complications and take stock of the model's few-shot ability, we take a leaf out of MT Bench's (Zheng et al., 2023) playbook, targeting only the final result that complies with the stipulated format. Model outputs bereft of matching outputs are branded as invalid. Given that the 5-shots prompt is near 2000 tokens and the context length of `Vicuna-33b-v1.3` and `falcon-40b-instruct` are limited to 2048 tokens, not all input may fit within the context length of these models, so we exclude the 5-shot results of these models.

## 4.3 CTF DATASET

### 4.3.1 OVERALL PERFORMANCE

We show the result of binary classification and multi-class classification for a single function with an averaged F1 score over raw decompiled code and manually reversed decompiled code in Figure 1. The results of the static analyzer and deep learning models are shown in Table 3 and Table 2. The full result of the CTF dataset is shown in Table 12.

Analysis of the CTF dataset revealed that GPT-4 maintained its acknowledged strong capability in various arenas—be it binary or multi-class classification—garnering impressive results. However, the Llama 2 family with RLHF Ouyang et al. (2022); Christiano et al. (2017) does not achieve projected expectations, suggesting that over-alignment will hurt the capability of the LLM OpenAI. When compared to the results of the supervised fine-tuned models of the same size, such as the Platypus or Vicuna, it is seen that these models demonstrated improved performance. Furthermore,

Table 2: Binary classification results in CTF and real-world datasets on deep-learning-based models.

| | CTF | | | Real-world | | |
|---|---|---|---|---|---|---|
| | F1 | Precision | Recall | F1 | Precision | Recall |
| VulBERTa | 0.354 | 0.350 | 0.391 | 0.406 | 0.456 | 0.388 |
| LineVul | 0.155 | 0.619 | 0.187 | 0.166 | 0.419 | 0.193 |
| CodeXGLUE | 0.375 | 0.341 | 0.617 | 0.429 | 0.437 | 0.462 |

Table 3: Multi-class classification results in CTF dataset on static analyzer tools, BinAbsInspector is provided with the binary, flawfinder and cppcheck is proviced with decompiled code.

| | Raw Decompiled code / Binary | | | Reversed Decompiled code | | |
|---|---|---|---|---|---|---|
| | F1 | Precision | Recall | F1 | Precision | Recall |
| flawfinder | 0.174 | 0.229 | 0.662 | 0.136 | 0.178 | 0.324 |
| cppcheck | 0.02 | 0.029 | 0.015 | 0.016 | 0.010 | 0.170 |
| BinAbsInspector | 0.604 | 0.652 | 0.563 | / | / | / |

Table 4: Multi-class classification results in MAGMA dataset on static analyzer tools, BinAbsInspector is provided with the binary, flawfinder, and cppcheck are provided with decompiled code and source code.

| | Raw Decompiled code / Binary | | | Source Code | | |
|---|---|---|---|---|---|---|
| | F1 | Precision | Recall | F1 | Precision | Recall |
| flawfinder | 0.064 | 0.043 | 0.191 | 0.052 | 0.035 | 0.369 |
| cppcheck | 0.069 | 0.055 | 0.093 | 0.096 | 0.068 | 0.185 |
| BinAbsInspector | 0.011 | 0.006 | 0.066 | / | / | / |

concerning different quantities of few-shot examples, there is a notable enhancement in the vulnerability mining capabilities of most models when comparing 5-shot to 2-shot scenarios.

When comparing with the baselines, static analyzers, and deep learning-based models, the GPT-3.5 and GPT-4 outperform the best baselines in terms of F1 in binary classification whereas the open-access models fall behind when compared to VulBERTa and CodeXGLUE. When comparing open-access models that are trained on identical datasets (eg. Llama 2, Vicuna), we notice that while the phenomenon of the scaling law persists, enhancements are notably limited. This indicates that merely augmenting the volume of parameters does not necessarily optimize a model's capacity for vulnerability detection adequately. Accordingly, complementing efforts need to be allocated to enhancing other components, notably the quality of the dataset used. Moreover, in the vulnerability detection domain, substantial disparities still exist between models comparable to GPT-3.5 and large-scale open-access alternatives, despite claims to the contrary.

### 4.3.2 ABLATION STUDY ON PROVIDED INFORMATION

We conduct a comparison of the model's capacity when provided with varying types of information. The lower figure in Figure 8 illustrates the outcome of providing all functions within a binary in the CTF challenge compared to only a single function is provided. Experimental data revealed that when provided with reverse-engineered pseudocode, several models, specifically GPT-4, GPT-3.5, Falcon-40b, and Vicuna, demonstrated improved performance, and a decline is less common. It suggests these models, much like humans, comprehend well-formatted and readable code better than merely decompiled code. Interestingly, when comparing performance on giving more context, GPT-4 showes stable results whether given a single function or the entire binary, while GPT-3.5's accuracy decreased when exposed to more functions, suggesting that GPT-4 handles additional context more effectively. Regarding open-access models, the majority of them have experienced performance improvements, indicating that given the context allows, acquiring more context can assist the model in making better judgments. This is true even for datasets like CTF, where vulnerabilities often only appear within a single function.

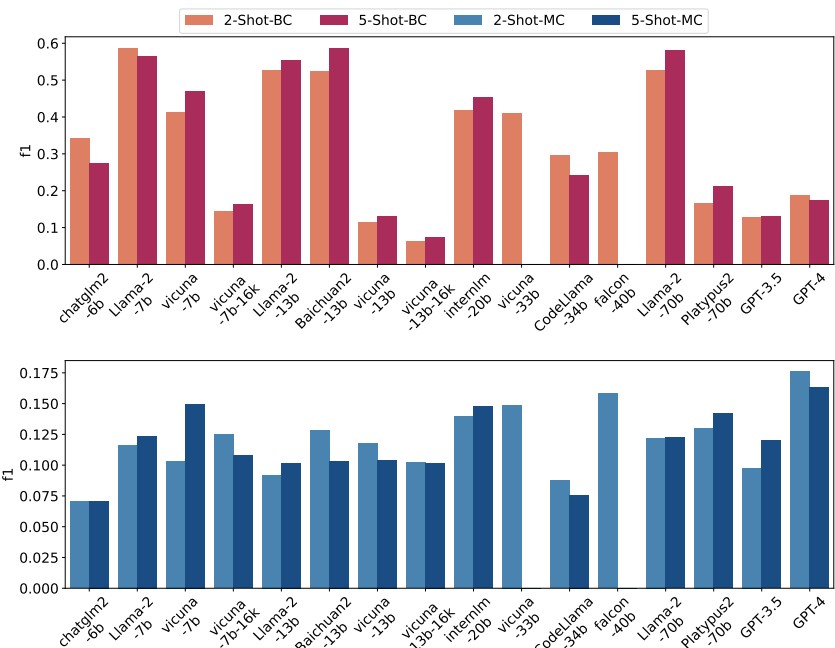

Figure 2: Few-shot conversation for binary classification (whether a function is vulnerable) and multi-class classification (which vulnerability does the function have) in the real-world dataset.

## 4.4 REAL-WORLD DATASET

Figure 2 showcases data from the real-world dataset as a collective. Along this, the results of the static analyzer and deep learning models are shown in Table 4 and Table 2. The full result on the real-world dataset is shown in Table 13.

As we delve into the challenging area of real-world vulnerability detection, it is evident that all LLMs, underperform when provided with just a single function. This is due to the fact that a comprehensive understanding of the entire program is needed. Despite this, GPT-4 outperforms other models in scenarios requiring multi-class classifications, where the requisite for high model capabilities is more pronounced. Conversely, performance declines in binary classification scenarios. But when compared with the deep learning models, the LLM performs relatively worse than them. Upon examining their predictions in Section 4.5 and H, we discern that their decreased proficiency often stems from the model's excessive conservatism—namely, its propensity to yield 'No Vulnerability'. This tendency contrasts with that of deep learning models, which, when trained on a specific dataset, strive to make judgments on the input content as far as possible. This may account for certain disparities between large-scale models and deep learning models.

When considering the single magma functions against those offering a more complete context (as depicted in the upper figure in Figure 9), there is negligible performance alteration. The LLMs may find it challenging to discern correctly due to the presence of a large amount of irrelevant normal coding, consequently undermining performance. The lower figure in Figure 9 extends the comparison, juxtaposing the magma source code with the decompiled code extracted using IDA — a comparison that yields similar results. This outcome is out of sync with those recorded in the CTF competition. We credit this discrepancy to the increased complexity found within the real-world vulnerability dataset as opposed to the CTF dataset.

## 4.5 ANALYSIS OF THE MODELS' PERFORMANCE

In Section H, we have presented and analyzed the performance of models on various datasets. It is noteworthy that despite the suboptimal performance of all models on real-world datasets, the reasons for their divergent outcomes are distinct. This includes a common tendency toward conservatism,

where the models are inclined to identify the absence of vulnerabilities. Notably, models derived from Reinforcement Learning from Human Feedback (RLHF) exhibit a strong bias towards certain types of vulnerabilities, highlighting that RLHF may align models more closely with human values or preferences, yet could also intensify certain biases. This could potentially conflict with RLHF's intent to minimize biases related to values or safety, rendering it less suitable for other tasks.

## 5 DISCUSSION

### 5.1 EVALUATION ON ROOT CAUSE DESCRIPTION

In this study, we advocate for a vulnerability dataset characterized by natural language. Given the abundance of results derived from numerous models, manual evaluation of every outcome is impractical. Further complicating matters, the GPT-4's limited comprehension of vulnerabilities restricts its utility as a proficient critic akin to MT Bench (Zheng et al., 2023), thereby undermining the evaluation of the model's results. We envisage an expansion of similar datasets, accompanied by the development of enhanced automated evaluation methodologies.

### 5.2 LIMITATION OF DECOMPILED CODE

We perform vulnerability detection for closed-source software based on decompiled code, but decompiled code itself also has certain limitations, and there will be errors and information loss, preventing the vulnerabilities from being reflected at the decompiled code level. We show a bad case of decompiled code in Section K. The decompilers must adapt to new compiler-generated assembly patterns (Basque et al.), which is a challenge beyond the scope of this study. Better yet, assembly code could be processed directly by an LLM, though its understanding falls short compared to the original decompiled code. Recent advances in effectively encoding assembly language (Pei et al., 2023; Wang et al., 2022) might offer a workaround, allowing an LLM to interpret assembly directly, similar to LLaVA (Liu et al., 2023a), bypassing the need for decompiled code.

### 5.3 FUTURE FOR VULNERABILITY DETECTION WITH LLM

We illustrate how GPT-4 noticeably surpasses other models on relatively simplistic datasets, such as CTF, indicative of its certain proficiency in vulnerability mining. Conversely, when considering the real-world dataset, none of the models, exhibited exceptional vulnerability mining prowess. The increasing complexity of software necessitates a deeper understanding to conduct effective vulnerability research. Enhancing our comprehension of entire projects (Bairi et al., 2023) is crucial to uncover more significant vulnerabilities. Additionally, tools like static analysis can support LLMs by providing supplementary data or aiding in challenging tasks (Schick et al., 2023; Yang et al., 2023a). Moreover, models can act as knowledge repositories, complementing other discovery techniques like fuzzing or static analysis (Li et al., 2023; Liu et al., 2023b).

## 6 CONCLUSION

In this study, we introduced a comprehensive vulnerability benchmark dataset, `VulBench`, and conducted an extensive evaluation of LLMs in the field of software vulnerability detection. Our research demonstrates that in certain scenarios, LLMs, particularly GPT-4, outperforms traditional deep learning-based models and static analyzers, especially in CTF datasets. This finding not only underscores the potential application of LLMs in enhancing software security but also opens new avenues for future research in automated vulnerability detection.

However, the performance of all models, including LLMs, drops in more complex real-world datasets. This suggests that while LLMs show promise in handling simplified scenarios, they still face challenges in understanding and analyzing complex software systems. Future research could focus on enhancing the ability of LLMs to process complex projects and explore how to synergize LLMs with other vulnerability detection techniques such as fuzzing or static analysis to maximize their strengths. In summary, this study paves new paths for understanding and utilizing LLMs to improve software security, providing valuable insights and directions for the advancement of automated vulnerability detection technologies.

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

## A  SELECTED LARGE LANGUAGE MODELS

Table 5 shows the selected LLMs and their corresponding model size. We use the model name on Huggingface.

| Name | Size |
|------|------|
| ChatGLM2-6b (Du et al., 2022; Zeng et al., 2023) | 6B |
| Llama-2-7b-chat-hf (Touvron et al., 2023) | |
| Vicuna-7b-v1.5 (Chiang et al., 2023) | 7B |
| Vicuna-7b-v1.5-16k | |
| Llama-2-13b-chat-hf | |
| Vicuna-13b-v1.5 | |
| Vicuna-13b-v1.5-16k | 13B |
| Baichuan2-13B-Chat (Baichuan, 2023) | |
| Internlm-20b-chat (Team, 2023) | 20B |
| Vicuna-33b-v1.3 | 33B |
| CodeLlama-34b-Instruct (Rozière et al., 2023) | 34B |
| Falcon-40b-instruct (Almazrouei et al., 2023) | 40B |
| Llama-2-70b-chat-hf | |
| Platypus2-70B-instruct (Lee et al., 2023) | 70B |
| GPT-3.5 (Ouyang et al., 2022) | |
| GPT-4 (OpenAI) | / |

Table 5: Names and sizes of the large models selected in the experiment, matched with their name on Huggingface.

## B  SELECTED BASELINES

We have selected three deep learning models and three rule-based static analysis tools according to Steenhoek et al. (2023); Lipp et al. (2022) to serve as baselines when comparing with the LLMs. The basic methods of the baselines are shown above.

- LineVul (Fu & Tantithamthavorn, 2022): CodeBERT-based model trained on the Devign dataset.
- CodeXGLUE (Lu et al., 2021): CodeBERT-based model trained on the Big-Vul dataset.
- VulBERTa (Hanif & Maffeis, 2022): Roberta-based model trained on the Devign dataset with custom tokenization pipeline.
- Cppcheck (Marjamäki, 2023): using a combination of syntactic and semantic analysis techniques to detect potential vulnerable C/C++ source code.
- Flawfinder (Wheeler, 2023): implements a syntactic analysis technique that scans C/C++ source code to report potential security flaws.
- BinAbsInspector (Keenlab, 2022): a static analyzer works on Ghidra's Pcode to scan vulnerabilities in binaries

## C  DATASET DETAILS

Table 6 shows the specific types of vulnerabilities contained in each dataset, and in multi-label classification, the label of each dataset is composed of the vulnerability types of the respective datasets and `No Vulnerability`. When we conduct binary classification experiments, our focus is function-level vulnerabilities, hence we break down complex datasets like CTF and MAGMA into individual functions for evaluation. For datasets like Devign, D2A, and Big-Vul, which are already segmented into functions, we directly input them for evaluation, without extra processing.

| Dataset | Types of Vulnerabilities |
|---------|--------------------------|
| CTF | buffer overflow, format string vulnerability, integer overflow, type confusion, use after free |
| MAGMA | buffer overflow, integer overflow, math error, memory leak, null pointer dereference, resource exhaustion, type confusion, use after free |
| Devign | buffer overflow, divide by zero, integer overflow, lack of sanity check, memory leak, null pointer dereference, race condition |
| Big-Vul | buffer overflow, improper control, integer overflow, math error, memory leak, null pointer dereference, race condition, resource exhaustion, use after free |
| D2A | buffer overflow, integer overflow, null pointer dereference, resource exhaustion |

Table 6: The types of vulnerabilities contained in each dataset.

## D    ILLUSTRATION OF THE NEED OF CONTEXT

We illustrate the need for context instead of only a single function is adequate in vulnerability detection in Listing 2, even if there are some patterns matched. `kbuf = kzalloc(len, GFP_KERNEL);` allocates a memory block. But it returns `-EFAULT` directly without freeing the `kbuf` if `copy_from_iter_full(kbuf, len, from)` returns 0. It is a common pattern for memory leak. But we still cannot determine whether it will lead to memory leak until we look inside the implement of the function `copy_from_iter_full` cause it may free the memory passed as parameters if anything goes wrong.

```
1  static ssize_t qrtr_tun_write_iter(struct kiocb *iocb, struct iov_iter *from)
2  {
3    struct file *filp = iocb->ki_filp;
4    struct qrtr_tun *tun = filp->private_data;
5    size_t len = iov_iter_count(from);
6    ssize_t ret;
7    void *kbuf;
8    kbuf = kzalloc(len, GFP_KERNEL);
9    if (!kbuf)
10     return -ENOMEM;
11   if (!copy_from_iter_full(kbuf, len, from))
12     return -EFAULT;
13   ret = qrtr_endpoint_post(&tun->ep, kbuf, len);
14   return ret < 0 ? ret : len;
15 }
```

Listing 2: Vulnerable funtion in CVE-2019-19079.

# E    EXAMPLE IN EACH DATASET

## E.1    CTF DATASET

In Listing 3, we give an example of raw decompiled code from IDA in a CTF binary. With the removal of symbol information, the initial global variables are now replaced with identifiers that are represented as addresses; for instance, dword_202050 and unk_202060. Likewise, any information regarding structures is removed. The revised way of accessing potential structure member variables now involves an address offset method, whose representative equivalent is *((_QWORD *)&unk_202070 + 4 * i). This differs from the previous, more common method which involved direct member name access.

```
1  int dword_202050;
2  char unk_202060[16];
3  char unk_202070[8];
4  _QWORD qword_202078[17];
5
6  unsigned __int64 sub_B0E()
7  {
8    unsigned int v1; // [rsp+0h] [rbp-10h] BYREF
9    unsigned int i; // [rsp+4h] [rbp-Ch]
10   unsigned __int64 v3; // [rsp+8h] [rbp-8h]
11
12   v3 = __readfsqword(0x28u);
13   if ( dword_202050 > 5 )
14   {
15     puts("You can't add any more notes!");
16     exit(0);
17   }
18   for ( i = 0; i <= 4 && *((_QWORD *)&unk_202070 + 4 * i); ++i )
19     ;
20   puts("How long is your note?");
21   _isoc99_scanf("%u", &v1);
22   if ( i > 0x80 )
23   {
24     puts("Too long!");
25     exit(0);
26   }
27   *((_QWORD *)&unk_202070 + 4 * i) = 1LL;
28   qword_202078[4 * i] = malloc(v1 + 1);
29   puts("Input your note name and note content:");
30   sub_AD6((char *)&unk_202060 + 32 * i, 21LL);
31   sub_AD6(qword_202078[4 * i], v1);
32   ++dword_202050;
33   puts("Done!");
34   return __readfsqword(0x28u) ^ v3;
35 }
```

Listing 3: A single function inside CTF problem t3sec2018_xueba. Decompiled code from IDA along with used global variables.

We restored the structure and renamed variables of the decompiled code shown in Listing 3, and the outcomes are showcased in Listing 4. Despite the manual reverse engineering effort, the control flow was not simplified. Moreover, certain patterns necessitated by compiler operations, such as the stack overflow canary check indicated by the code line v2 = __readfsqword(0x28u);, have been preserved. The decompiled code references library function names typically used internally by compilers, such as _isoc99_scanf, rather than their more common equivalents, like scanf. However, the structural reverse engineering has clarified that multiple identifiers - unk_202060, unk_202070, and qword_202078 - from Listing 3 are, in fact, references to the same structured variable Note notes[5]. The IDA, which utilizes access patterns to deduce structure, can not merge these identifiers. This reverse engineering procedure addresses this problem, enhancing the comprehensibility of the program's functioning.

Given a more understandable decompile, we can determine the vulnerability inside this function, and this will serve as the natural language description of the vulnerability. It is shown in Table 7

```
1  struct Note
2  {
3    char name[16];
4    __int64 used;
5    char *content;
6  };
7
8  Note notes[5];
9
10 void add_note()
11 {
12   int v0;
13   unsigned int i;
14   unsigned __int64 v2;
15
16   v2 = __readfsqword(0x28u);
17   if ( note_count > 5 )
18   {
19     puts("You can't add any more notes!");
20     exit(0);
21   }
22   for ( i = 0; i <= 4 && notes[i].used; ++i )
23     ;
24   puts("How long is your note?");
25   _isoc99_scanf("%u", &v0);
26   if ( i > 0x80 )
27   {
28     puts("Too long!");
29     exit(0);
30   }
31   notes[i].used = 1LL;
32   notes[i].content = (char *)malloc((unsigned int)(v0 + 1));
33   puts("Input your note name and note content:");
34   read_str(notes[i].name, 0x15u);
35   read_str(notes[i].content, v0);
36   ++note_count;
37   puts("Done!");
38 }
```

Listing 4: A single function inside CTF problem t3sec2018_xueba. Manually reversed decompiled code from IDA along with used global variables.

the name in the Node struct has only a size of 16. But in the statement read_str(notes[i].name, 0x15u);, the attacker can feed at most 0x15 bytes into notes[i].name, leading to a Buffer-Overflow vulnerability.

Table 7: The natural language description of vulnerability inside Listing 4.

## E.2 MAGMA DATASET

```
1  static int wav_write_header(SF_PRIVATE *psf, int calc_length) {
2      ...
3  #ifdef MAGMA_ENABLE_FIXES
4      /* Make sure we don't read past the loops array end. */
5      if (psf->instrument->loop_count > ARRAY_LEN(psf->instrument->loops))
6          psf->instrument->loop_count = ARRAY_LEN(psf->instrument->loops);
7  #endif
8  #ifdef MAGMA_ENABLE_CANARIES
9      MAGMA_LOG("%MAGMA_BUG%",
10               apsf->instrument->loop_count > ARRAY_LEN(psf->instrument->loops));
11 #endif
12     for (tmp = 0; tmp < psf->instrument->loop_count; tmp++) {
13         int type;
14         type = psf->instrument->loops[tmp].mode;
15         ...
16     }
17     ...
18 }
```

Listing 5: MAGMA example (SND025) with an 'Out-of-Bound-Read' Vulnerability.

In Listing 5, we show a case where we can use the 'canary' and the corresponding fixes to help understand the root cause of the vulnerability in the MAGMA dataset. In each MAGMA bug, the corresponding source code contains two marco MAGMA_ENABLE_CANARIES and MAGMA_ENABLE_FIXES to check whether the bug is triggered and the fixes for the bug. The code snippets inside marco MAGMA_ENABLE_CANARIES will act as the 'canary' for checking whether the vulnerability is triggered during Fuzzing. The code inside marco MAGMA_ENABLE_FIXES will serve as the fix for this vulnerability. In this example, we identify the flaw as an 'Out-of-Bound-Read' according to the canary `psf->instrument->loop_count > ARRAY_LEN(psf->instrument->loops)` and correlate with subsequent code, that is `psf->instrument->loops[tmp].mode`.

### E.3 BIG-VUL DATASET

In Listing 6, we show an example from Big-Vul dataset to demonstrate how we normalize the 'CWE ID', 'Summary', and 'Vulnerability Classification' into the label of multi-class classification. In the original Big-Vul dataset, there are 'CWE ID', 'Summary', and 'Vulnerability Classification' to describe the vulnerability of the function, however, they do not always exist. For example, in the vulnerable function in Listing 6, it contains 'CWE ID' (CWE-416) and 'Summary' (Shown in Table 8 but the 'Vulnerability Classification' is absent. Then we can determine that there is a 'Use-After-Free' vulnerability in this function.

```c
int blk_init_allocated_queue(struct request_queue *q)
{
  WARN_ON_ONCE(q->mq_ops);

  q->fq = blk_alloc_flush_queue(q, NUMA_NO_NODE, q->cmd_size);
  if (!q->fq)
    return -ENOMEM;

  if (q->init_rq_fn && q->init_rq_fn(q, q->fq->flush_rq, GFP_KERNEL))
    goto out_free_flush_queue;

  if (blk_init_rl(&q->root_rl, q, GFP_KERNEL))
    goto out_exit_flush_rq;

  INIT_WORK(&q->timeout_work, blk_timeout_work);
  q->queue_flags    |= QUEUE_FLAG_DEFAULT;

  /*
   * This also sets hw/phys segments, boundary and size
   */
  blk_queue_make_request(q, blk_queue_bio);

  q->sg_reserved_size = INT_MAX;

  if (elevator_init(q))
    goto out_exit_flush_rq;
  return 0;

out_exit_flush_rq:
  if (q->exit_rq_fn)
      q->exit_rq_fn(q, q->fq->flush_rq);
out_free_flush_queue:
  blk_free_flush_queue(q->fq);
  return -ENOMEM;
}
```

Listing 6: A Big-Vul example (CVE-2018-20856) containing a Use-After-Free vulnerability.

An issue was discovered in the Linux kernel before 4.18.7. In `block/blk-core.c`, there is an `__blk_drain_queue()` use-after-free because a certain error case is mishandled.

Table 8: Summary of Listing 6 in Big-Vul dataset

# F CASE STUDY

Here we show some cases (1 CTF and 2 real-world cases) where GPT-4 understand the code correctly and successfully detect the vulnerabilities. The vulnerable functions are shown in Listing 7, Listing 8 and Listing 10, and the corresponding answers of GPT-4 are shown in Table 9, Table 10, and Table 11.

## F.1 CTF CASE 1

In CTF problem `huxiangbei_2019_namesystem` shown in Listing 7 and corresponding answer in Table 9, GPT-4 successfully identify `a1[v0] = a1[v0 + 1]` will cause two pointers inside the `a1` point to the same memory, resulting the possible Use-After-Free vulnerability.

```c
void __fastcall delete_name()
{
  unsigned int v0; // [rsp+Ch] [rbp-4h]

  printf("The id you want to delete:");
  v0 = read_int();
  if ( v0 <= 0x13 && a1[v0] )
  {
    free(a1[v0]);
    a1[v0] = 0LL;
    while ( (int)(v0 + 1) <= 19 )
    {
      a1[v0] = a1[v0 + 1];
      ++v0;
    }
    puts("Done!");
  }
  else
  {
    puts("Invalid id !!");
  }
}
```

Listing 7: Function delete_name in CTF challenge huxiangbei_2019_namesystem.

Output:
The code has a potential Use-After-Free vulnerability. The vulnerability occurs in the delete_name() function:
void __fastcall delete_name()
{
   ...
   free(a1[v0]);
   a1[v0] = 0LL;
   while ( (int)(v0 + 1) ≤ 19 )
   {
      a1[v0] = a1[v0 + 1];
      ++v0;
   }
   ...
}
After freeing the memory at a1[v0], the function shifts a1[v0 + 1] to a1[v0] without checking if a1[v0 + 1] is NULL. This could lead to a use-after-free vulnerability when the program accesses the freed memory in subsequent operations.

TYPE: Use-After-Free

Table 9: GPT-4's Answer for Listing 7 in Multi-class Classification.

## F.2 REAL WORLD CASE 1

In function from real-world program 'tcpdump' shown in Listing 8, the LLM identifies the difference between the first call to l_strnstart and the subsequent l_strnstart, and report that as a possible vulnerability. However, to verify the exploitability of this vulnerability, we need to examine the function l_strnstart. The developer may check the length inside l_strnstart as shown in Listing 9.

```
1  beep_print(netdissect_options *ndo, const u_char *bp, u_int length)
2  {
3
4  if (l_strnstart("MSG", 4, (const char *)bp, length)) /* A REQuest */
5    ND_PRINT((ndo, " BEEP MSG"));
6  else if (l_strnstart("RPY ", 4, (const char *)bp, length))
7    ND_PRINT((ndo, " BEEP RPY"));
8  else if (l_strnstart("ERR ", 4, (const char *)bp, length))
9    ND_PRINT((ndo, " BEEP ERR"));
10 else if (l_strnstart("ANS ", 4, (const char *)bp, length))
11   ND_PRINT((ndo, " BEEP ANS"));
12 else if (l_strnstart("NUL ", 4, (const char *)bp, length))
13   ND_PRINT((ndo, " BEEP NUL"));
14 else if (l_strnstart("SEQ ", 4, (const char *)bp, length))
15   ND_PRINT((ndo, " BEEP SEQ"));
16 else if (l_strnstart("END", 4, (const char *)bp, length))
17   ND_PRINT((ndo, " BEEP END"));
18  else
19   ND_PRINT((ndo, " BEEP (payload or undecoded)"));
20 }
```

Listing 8: A real-world function named beep_print.

Output:
The function has a buffer overflow vulnerability due to the code:
```
if (l_strnstart("MSG", 3, (const char *)bp, length)) /* A REQuest */
   ND_PRINT((ndo, " BEEP MSG"));
```

The issue lies in the second argument of l_strnstart() function, which should be 3 instead of 4. This is because the string "MSG" has a length of 3, not 4. If the length of bp is less than 3, the function will read beyond the buffer boundary, leading to a buffer overflow vulnerability.

TYPE: Buffer-Overflow

Table 10: GPT-4's Answer for Listing 8 in Multi-class Classification.

```
1  static int
2  l_strnstart(netdissect_options *ndo, const char *tstr1, u_int tl1,
3      const char *str2, u_int l2)
4  {
5      if (!ND_TTEST_LEN(str2, tl1)) {
6      /* We don't have tl1 bytes worth of captured data
7       * for the string, so we can't check for this
8       * string. */
9      return 0;
10   }
11   if (tl1 > l2)
12      return 0;
13   return (strncmp(tstr1, str2, tl1) == 0 ? 1 : 0);
14 }
```

Listing 9: l_strnstart function called inside beep_print. The commit (guyharris) to fix this CVE did not modify how the l_strnstart is called, instead, the developer add an extra check on length.

### F.3 REAL WORLD CASE 2

In function from real-world program V8 shown in Listing 10, GPT-4 uses its knowledge to identify that `context_->CallFunction` will potentially retain a reference to the object, yielding a Use-After-Free vulnerability. This requires that the LLM has internal knowledge about the side effects of the calling function. However, when dealing with a new project, it requires that the called function is present in the context as the LLM does not know the project.

```cpp
void CallbackAndDie(bool succeeded) {
    v8::Isolate* isolate = context_->isolate();
    v8::HandleScope handle_scope(isolate);
    v8::Local<v8::Value> args[] = {v8::Boolean::New(isolate, succeeded)};
    context_->CallFunction(v8::Local<v8::Function>::New(isolate, callback_),
                           arraysize(args), args);
     delete this;
}
```

Listing 10: A real-world function named `CallbackAndDie`.

> Output:
> The code snippet appears to be a callback function for a V8 JavaScript engine context. It creates a handle scope and calls a function with the given arguments. After the function call, the object is deleted. There is a potential Use-After-Free vulnerability if the callback function retains a reference to the object and attempts to access it after the "delete this" statement.
>
> TYPE: Use-After-Free

Table 11: GPT-4's Answer for Listing 10 in Multi-class Classification.

### F.4 REAL WORLD CASE 3

In the function from a real-world Lua program shown in Listing 11, all LLMs fail to identify the integer overflow vulnerability. This vulnerability is still challenging for experienced security researchers, which requires LLMs to understand how integers are stored in computers, and to realize that the logic of this code segment fails to consider the only corner case: the result of taking the negation of INT_MIN is still INT_MIN. When the `findvararg` function is called, if n equals INT_MIN, it leads to integer overflow in the expression `*pos=ci->func - nextra + (n-1)`.

```c
static const char *findvararg(CallInfo *ci, int n, StkId *pos) {
  if (clLvalue(s2v(ci->func))->p->is_vararg) {
    int nextra = ci->u.l.nextraargs;
    if (n <= nextra) {
      *pos = ci->func - nextra + (n - 1);
      return "(vararg)"; /* generic name for any vararg */
    }
  }
  return NULL; /* no such vararg */
}

const char *luaG_findlocal(lua_State *L, CallInfo *ci, int n, StkId *pos) {
  StkId base = ci->func + 1;
  const char *name = NULL;
  if (isLua(ci)) {
    if (n < 0) /* access to vararg values? */
      return findvararg(ci, -n, pos);
    else
      name = luaF_getlocalname(ci_func(ci)->p, n, currentpc(ci));
  }
  /* 12 lines here */
  if (pos)
    *pos = base + (n - 1);
  return name;
}
```

Listing 11: A real-world integer overflow vulnerability in lua.

# G ILLUSTRATION OF FEW SHOT LEARNING CONVERSATION

To ease the parsing process, we use a few shot learning to teach the models how to answer the question in a predefined format, especially those models not aligned to follow human instruction well, such as Baichuan2, internlm, and falcon. We show how we format the 2-shot conversation template in Figure 3 and Figure 4. In each few-shot conversation, we teach the model to follow the format as VULNERABLE: NO, VULNERABLE: YES or TYPE: [type].

Please note that the following codes and corresponding example answers for only for illustration.

```c
Please review and analyse the following code, determine whether it is vulnerable.
Think step by step. Answer as "VULNERABLE: YES/NO".
beep_print(netdissect_options *ndo, const u_char *bp, u_int length)
{
        if (l_strnstart("MSG", 4, (const char *)bp, length)) /* A REQuest */
        ND_PRINT((ndo, " BEEP MSG"));
else if (l_strnstart("RPY ", 4, (const char *)bp, length))
        ND_PRINT((ndo, " BEEP RPY"));
else if (l_strnstart("ERR ", 4, (const char *)bp, length))
        ND_PRINT((ndo, " BEEP ERR"));
else if (l_strnstart("ANS ", 4, (const char *)bp, length))
        ND_PRINT((ndo, " BEEP ANS"));
else if (l_strnstart("NUL ", 4, (const char *)bp, length))
        ND_PRINT((ndo, " BEEP NUL"));
else if (l_strnstart("SEQ ", 4, (const char *)bp, length))
        ND_PRINT((ndo, " BEEP SEQ"));
else if (l_strnstart("END", 4, (const char *)bp, length))
        ND_PRINT((ndo, " BEEP END"));
    else
        ND_PRINT((ndo, " BEEP (payload or undecoded)"));
}
```

```
The function has a buffer overflow vulnerability due to the code:
if (l_strnstart("MSG", 4, (const char *)bp, length)) /* A REQuest */
        ND_PRINT((ndo, " BEEP MSG"));
If the length of bp is less than 4, the function will read beyond the buffer
boundary, leading to a buffer overflow vulnerability.

VULNERABLE: YES
```

```c
Please review and analyse the following code, determine whether it is vulnerable.
Think step by step. Answer as "VULNERABLE: YES/NO".
static int announce_self_create(uint8_t *buf, uint8_t *mac_addr)
{
    uint32_t magic = EXPERIMENTAL_MAGIC;
    uint16_t proto = htons(ETH_P_EXPERIMENTAL);
    /* FIXME: should we send a different packet (arp/rarp/ping)? */
    memset(buf, 0, 64);
    memset(buf, 0xff, 6);          /* h_dst */
    memcpy(buf + 6, mac_addr, 6); /* h_src */
    memcpy(buf + 12, &proto, 2);  /* h_proto */
    memcpy(buf + 14, &magic, 4);  /* magic */
    return 64; /* len */
}
```

```
The memset function is used to initially set the entire buffer to zero, ensuring
there's no lingering junk data. The function doesn't take any untrusted inputs from
the outside. The only input is mac_addr, and we assume the code calling this
function ensures it's a valid MAC address. There is no apparently vulnerability in
the function.

VULNERABLE: NO
```

Few-shot Conversation
-----------------------------------------------------------------------------------------------------
User Conversation

```
Please review and analyse the following code, determine whether it is vulnerable.
Think step by step. Answer as "VULNERABLE: YES/NO".
[Code to be evaluated]
```

Figure 3: Few-shot conversation for binary classification (whether a function is vulnerable). Texts in the green box are the queries, and texts in the yellow box are the model's answers.

```
Please review and analyse the following code, determine whether it has vulnerabilities in the
following type list.
[Possible Types of Vulnerabilities]
Think step by step. Answer as [Your Explanation] TYPE: [Vulnerability TYPE].
beep_print(netdissect_options *ndo, const u_char *bp, u_int length)
{
        if (l_strnstart("MSG", 4, (const char *)bp, length)) /* A REQuest */
        ND_PRINT((ndo, " BEEP MSG"));
else if (l_strnstart("RPY ", 4, (const char *)bp, length))
        ND_PRINT((ndo, " BEEP RPY"));
else if (l_strnstart("ERR ", 4, (const char *)bp, length))
        ND_PRINT((ndo, " BEEP ERR"));
else if (l_strnstart("ANS ", 4, (const char *)bp, length))
        ND_PRINT((ndo, " BEEP ANS"));
else if (l_strnstart("NUL ", 4, (const char *)bp, length))
        ND_PRINT((ndo, " BEEP NUL"));
else if (l_strnstart("SEQ ", 4, (const char *)bp, length))
        ND_PRINT((ndo, " BEEP SEQ"));
else if (l_strnstart("END", 4, (const char *)bp, length))
        ND_PRINT((ndo, " BEEP END"));
    else
        ND_PRINT((ndo, " BEEP (payload or undecoded)"));
}
```

```
The function has a buffer overflow vulnerability due to the code:
if (l_strnstart("MSG", 4, (const char *)bp, length)) /* A REQuest */
        ND_PRINT((ndo, " BEEP MSG"));
If the length of bp is less than 4, the function will read beyond the buffer
boundary, leading to a buffer overflow vulnerability.

TYPE: Buffer-Overflow
```

```
Please review and analyse the following code, determine whether it has vulnerabilities in the
following type list.
[Possible Types of Vulnerabilities]
Think step by step. Answer as [Your Explanation] TYPE: [Vulnerability TYPE].
static int announce_self_create(uint8_t *buf, uint8_t *mac_addr)
{
    uint32_t magic = EXPERIMENTAL_MAGIC;
    uint16_t proto = htons(ETH_P_EXPERIMENTAL);
    /* FIXME: should we send a different packet (arp/rarp/ping)? */
    memset(buf, 0, 64);
    memset(buf, 0xff, 6);          /* h_dst */
    memcpy(buf + 6, mac_addr, 6);  /* h_src */
    memcpy(buf + 12, &proto, 2);   /* h_proto */
    memcpy(buf + 14, &magic, 4);   /* magic */
    return 64; /* len */
}
```

```
The memset function is used to initially set the entire buffer to zero, ensuring
there's no lingering junk data. The function doesn't take any untrusted inputs from
the outside. The only input is mac_addr, and we assume the code calling this
function ensures it's a valid MAC address. There is no apparently vulnerability in
the function.
TYPE: No-Vulnerability
```

Few-shot Conversation
------------------------------------------------------------------------------------------------
User Conversation

```
Please review and analyse the following code, determine whether it has vulnerabilities in the
following type list.
[Possible Types of Vulnerabilities]
Think step by step. Answer as [Your Explanation] TYPE: [Vulnerability TYPE].
[Code to be evaluated]
```

Figure 4: Few-shot conversation for multi-class classification (determine a function's vulnerability type). Texts in the green box are the queries, and texts in the yellow box are the model's answers.

# H  ANALYSIS OF THE MODELS' PERFORMANCE

## H.1  CONFIGUATION

In Figures 5, 6, and 7, we present the confusion matrices for select representative models. Within these matrices, each row enumerates the percentage of accurate or inaccurate classification of a specific vulnerability type against each predicted category. We demonstrate the results on the CTF, Big-Vul, and MAGMA datasets. As for the CTF and MAGMA datasets, we mix the 'single' function result and 'all' functions result. In the table, the row labeled 'No Vulns' indicates that the function is devoid of vulnerabilities. The column titled 'Invalid' quantifies the percentage wherein the model outputs results that are incompatible with the expected template outlined in Section 4.2.

## H.2  MODELS'S BEHAVIOUR

Upon analyzing the CTF dataset, as shown in Figure 5, both GPT-4 and GPT-3.5 predominantly yield correct responses with GPT-4 slightly ahead. However, GPT-3.5 tends to answer 'No Vulnerability' in the MAGMA and Big-Vul datasets, respectively. GPT-4's performance, while cautious, is marred by a tendency to incorrectly flag potential 'Null Pointer Dereference' and 'Memory Leak' issues due to the lack of additional context, resulting in hypersensitivity and the misconception.

The Llama 2 models, despite using similar datasets for pretraining and the same RLHF methods for alignment, display starkly different behaviors across datasets. The 70B consistently predicts 'Buffer Overflow' on the CTF dataset, while the 13B variant exhibits a random pattern in its responses, showing a marginal tendency towards 'Use After Free'. This bias is more evident on the Big-Vul and MAGMA datasets, which is also replicated in the Baichuan2 and Internlm models.

Meanwhile, regarding the Vicuna and Platypus models, which are fine-tuned on the Supervised Fine-Tuning (SFT) datasets, there is a greater propensity to output 'No Vulnerability' when uncertain. Platypus, in particular, despite leveraging a substantial corpus of GPT-4 generated data, fails to reach the level of GPT-4, exhibiting behaviors that are not entirely consistent with GPT-4 but are noticeably conservative, frequently returning 'No Vulnerability'.

## H.3  CAPABILITIES ON DIFFERENT VULNERABILITIES

In conducting a specific analysis of a model's capacity for different kinds of vulnerabilities, it has been observed that in CTF, the model exhibits optimal performance in identifying 'Integer Overflow' vulnerability, almost always recognizing them correctly. Simultaneously, the model is relatively proficient in handling 'Format String' vulnerability. Apart from the models having a significant bias, most models struggle with type confusion during CTF encounters. This may be in part because the models are pre-trained with considerably fewer examples of type confusion, and also because type confusion often occurs in multiple functions, which place higher demands on the model's capability.

However, in the MAGMA and Big-Vul datasets, Platypus2, which utilizes significant training corpus derived from GPT-4, demonstrate certain detection capabilities for Buffer Overflow, possibly because examples of 'Buffer Overflow' are more common in the standard internet corpus. Nevertheless, these models do not perform as well with other types of vulnerabilities. Surprisingly, Llama 2 70B, which has a tendency to output 'Buffer Overflow', shows a remarkably good result in detecting 'Race Conditions' in the Big-Vul dataset, presenting an intriguing phenomenon.

## H.4  MODEL BIAS

In response to the considerable biases toward certain vulnerabilities observed across various models, we can discern that these biases—manifest in models such as Llama 2 Chat, Baichuan2, and InternLM—are likely interconnected with the RLHF. During the RLHF process, it's conceivable that models might develop a propensity for outputs that align more with human values or preferences rather than reflecting the ultimate truth—especially when they lack the capability to independently evaluate the flaws. This tendency to err towards human-like prejudice when presenting a vulnerability is evident. Conversely, models derived through SFT exhibit significantly another biases, tending to be comparatively conservative.

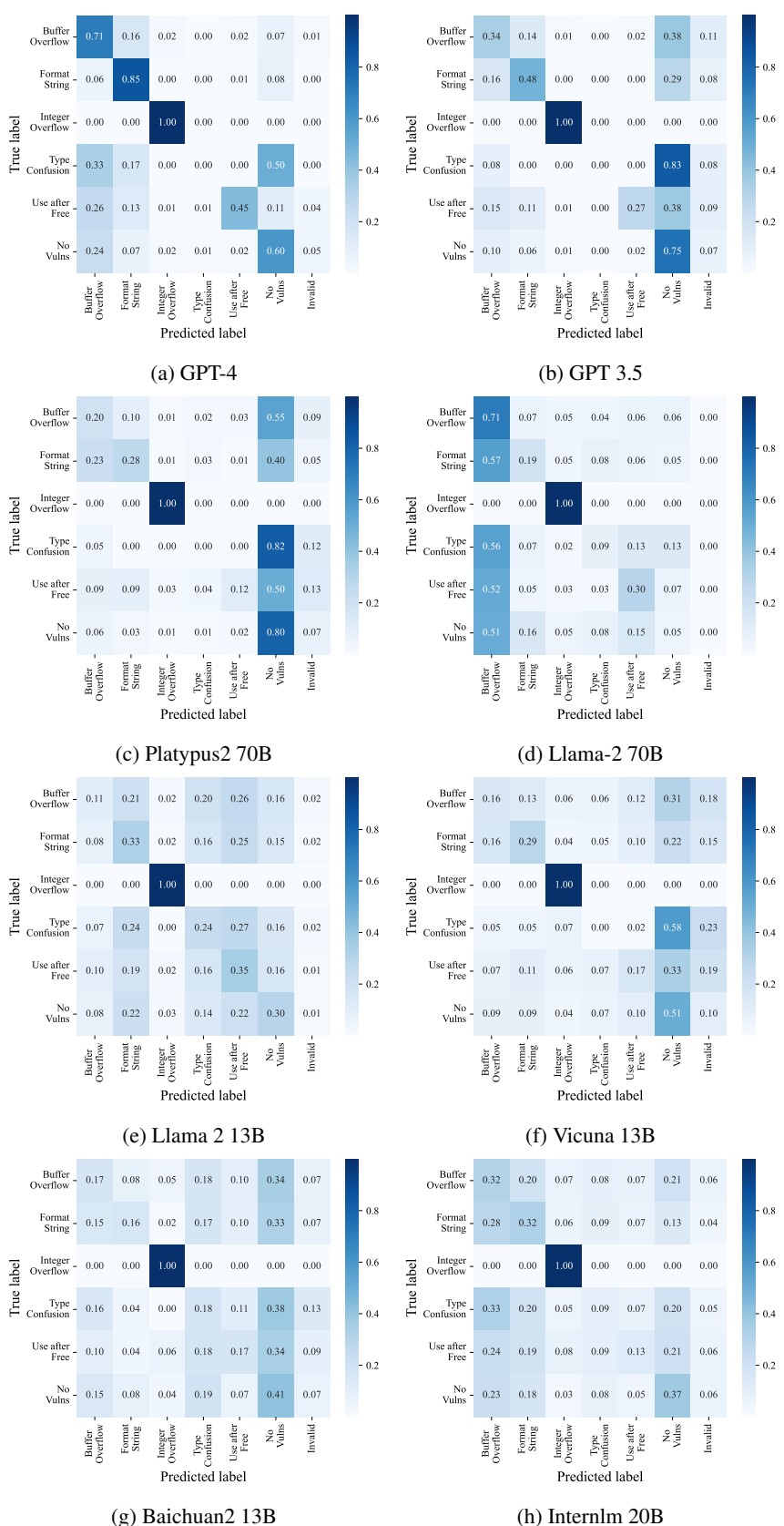

Figure 5: Confusion matrix of different models on the CTF dataset

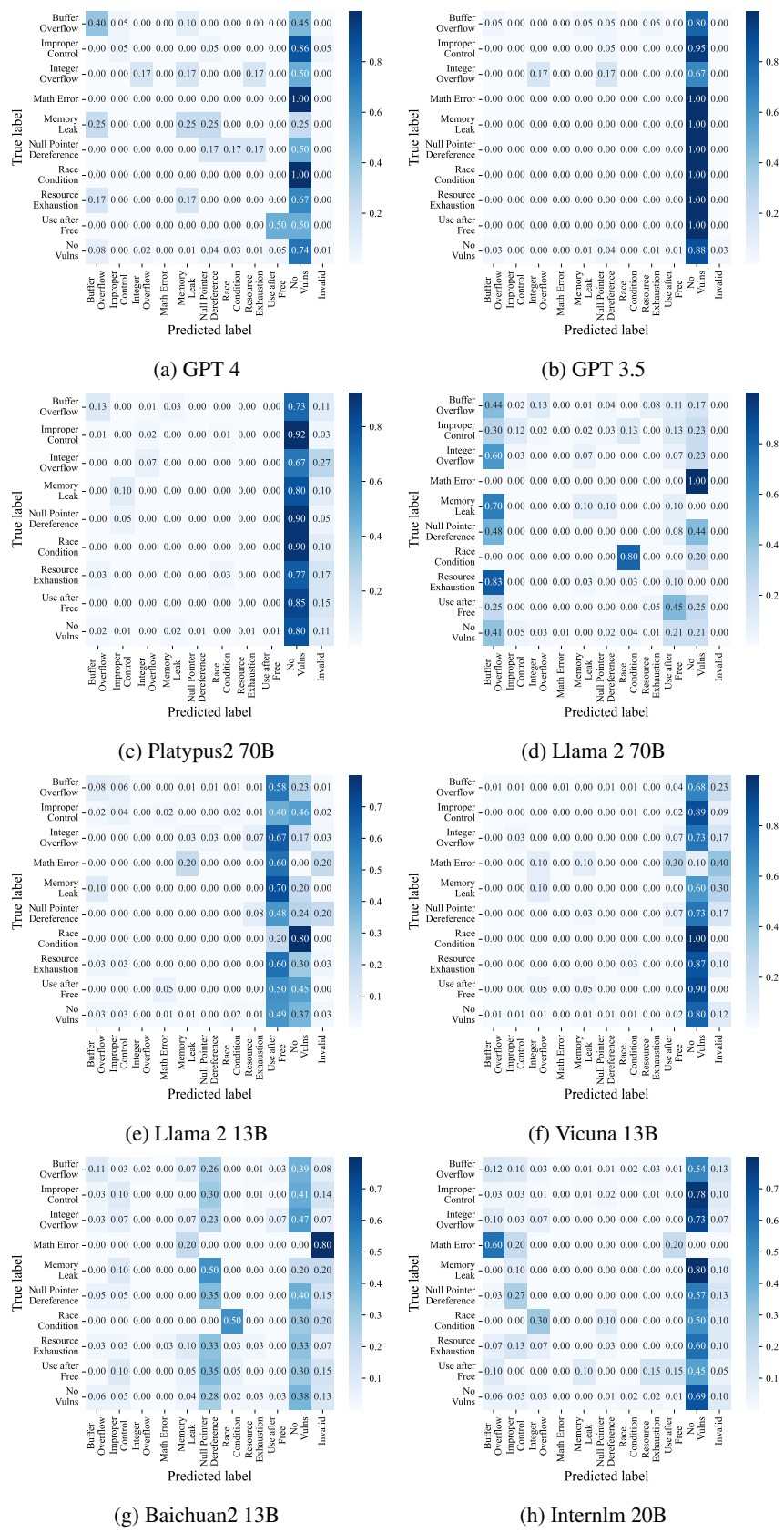

Figure 6: Confusion matrix of different models on the Big-Vul dataset

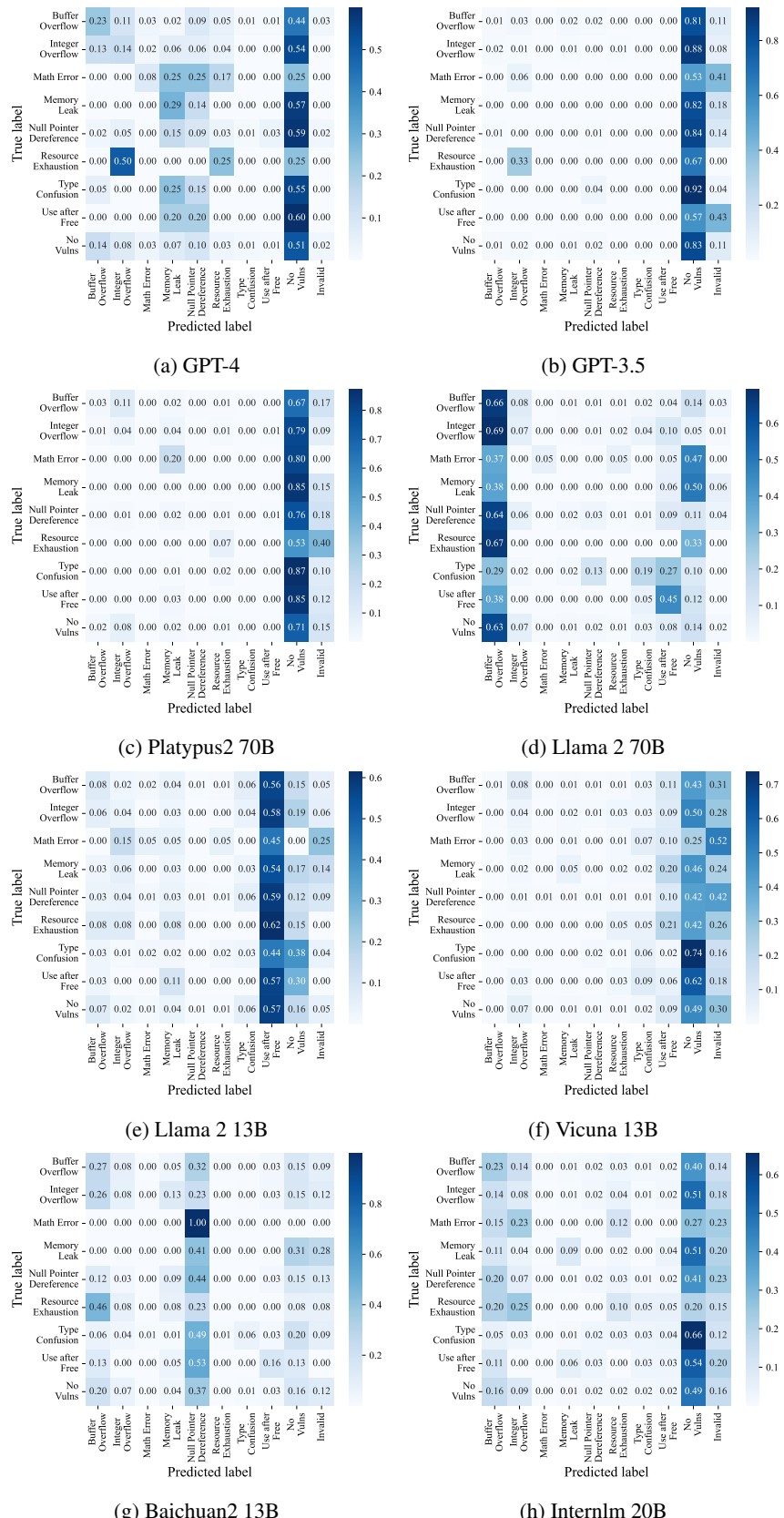

Figure 7: Confusion matrix of different models on the MAGMA dataset

# I   FULL EVALUATION RESULT

We show the full evaluation result in Table 12 and Table 13.

Table 12: Full evaluation results on CTF dataset

| Size | Model | Type | Binary Classification | | | Multi-class Classification | | |
|---|---|---|---|---|---|---|---|---|
| | | | F1 | Precision | Recall | F1 | Precision | Recall |
| 6B | ChatGLM2 | 2 Shot | 0.281 | 0.177 | 0.672 | 0.103 | 0.182 | 0.237 |
| | | 5 Shot | 0.255 | 0.168 | 0.529 | 0.081 | 0.181 | 0.135 |
| 7B | Llama-2 | 2 Shot | 0.291 | 0.171 | 0.966 | 0.104 | 0.170 | 0.179 |
| | | 5 Shot | 0.288 | 0.171 | 0.917 | 0.099 | 0.172 | 0.190 |
| | Vicuna | 2 Shot | 0.310 | 0.188 | 0.882 | 0.135 | 0.195 | 0.302 |
| | | 5 Shot | 0.325 | 0.198 | 0.902 | 0.128 | 0.203 | 0.282 |
| | Vicuna-16k | 2 Shot | 0.331 | 0.234 | 0.567 | 0.164 | 0.212 | 0.301 |
| | | 5 Shot | 0.335 | 0.232 | 0.601 | 0.147 | 0.208 | 0.215 |
| 13B | Llama-2 | 2 Shot | 0.291 | 0.172 | 0.948 | 0.133 | 0.184 | 0.279 |
| | | 5 Shot | 0.302 | 0.181 | 0.916 | 0.103 | 0.187 | 0.192 |
| | Baichuan2 | 2 Shot | 0.294 | 0.178 | 0.845 | 0.189 | 0.202 | 0.198 |
| | | 5 Shot | 0.289 | 0.170 | 0.972 | 0.068 | 0.186 | 0.217 |
| | Vicuna | 2 Shot | 0.336 | 0.246 | 0.530 | 0.188 | 0.203 | 0.208 |
| | | 5 Shot | 0.359 | 0.257 | 0.592 | 0.145 | 0.210 | 0.173 |
| | Vicuna-16k | 2 Shot | 0.329 | 0.236 | 0.545 | 0.169 | 0.195 | 0.226 |
| | | 5 Shot | 0.336 | 0.238 | 0.571 | 0.167 | 0.202 | 0.211 |
| 20B | InternLM | 2 Shot | 0.304 | 0.190 | 0.762 | 0.158 | 0.188 | 0.206 |
| | | 5 Shot | 0.325 | 0.206 | 0.768 | 0.142 | 0.202 | 0.242 |
| 33B | Vicuna | 2 Shot | 0.333 | 0.213 | 0.768 | 0.168 | 0.205 | 0.267 |
| | | 5 Shot | / | / | / | / | / | / |
| 34B | CodeLlama | 2 Shot | 0.319 | 0.224 | 0.553 | 0.148 | 0.207 | 0.226 |
| | | 5 Shot | 0.340 | 0.239 | 0.590 | 0.120 | 0.214 | 0.287 |
| 40B | Falcon | 2 Shot | 0.318 | 0.208 | 0.675 | 0.193 | 0.206 | 0.262 |
| | | 5 Shot | / | / | / | / | / | / |
| 70B | Llama-2 | 2 Shot | 0.295 | 0.173 | 0.993 | 0.100 | 0.197 | 0.340 |
| | | 5 Shot | 0.309 | 0.183 | 0.989 | 0.098 | 0.196 | 0.321 |
| | Platypus | 2 Shot | 0.352 | 0.289 | 0.455 | 0.234 | 0.246 | 0.236 |
| | | 5 Shot | 0.364 | 0.288 | 0.504 | 0.228 | 0.248 | 0.221 |
| / | GPT-3.5 | 2 Shot | 0.440 | 0.327 | 0.689 | 0.305 | 0.307 | 0.381 |
| | | 5 Shot | 0.429 | 0.340 | 0.590 | 0.296 | 0.305 | 0.308 |
| | GPT-4 | 2 Shot | 0.466 | 0.319 | 0.861 | 0.296 | 0.300 | 0.439 |
| | | 5 Shot | 0.483 | 0.356 | 0.750 | 0.379 | 0.357 | 0.489 |

Table 13: Full evaluation results on the real-world dataset

| Size | Model | Type | Binary Classification | | | Multi-class Classification | | |
|---|---|---|---|---|---|---|---|---|
| | | | F1 | Precision | Recall | F1 | Precision | Recall |
| 6B | ChatGLM2-6b | 2 Shot | 0.344 | 0.455 | 0.301 | 0.063 | 0.125 | 0.060 |
| | | 5 Shot | 0.274 | 0.456 | 0.218 | 0.065 | 0.134 | 0.073 |
| 7B | Llama-2 | 2 Shot | 0.588 | 0.463 | 0.891 | 0.106 | 0.150 | 0.124 |
| | | 5 Shot | 0.566 | 0.458 | 0.823 | 0.115 | 0.156 | 0.139 |
| | Vicuna | 2 Shot | 0.413 | 0.478 | 0.382 | 0.111 | 0.171 | 0.113 |
| | | 5 Shot | 0.470 | 0.465 | 0.495 | 0.138 | 0.201 | 0.129 |
| | Vicuna-16k | 2 Shot | 0.144 | 0.454 | 0.088 | 0.109 | 0.148 | 0.101 |
| | | 5 Shot | 0.165 | 0.457 | 0.101 | 0.095 | 0.141 | 0.079 |
| 13B | Llama-2 | 2 Shot | 0.527 | 0.453 | 0.706 | 0.083 | 0.140 | 0.110 |
| | | 5 Shot | 0.555 | 0.475 | 0.725 | 0.089 | 0.174 | 0.150 |
| | Baichuan2 | 2 Shot | 0.525 | 0.467 | 0.659 | 0.124 | 0.157 | 0.130 |
| | | 5 Shot | 0.588 | 0.450 | 0.936 | 0.101 | 0.188 | 0.168 |
| | Vicuna | 2 Shot | 0.114 | 0.476 | 0.067 | 0.107 | 0.143 | 0.116 |
| | | 5 Shot | 0.133 | 0.510 | 0.080 | 0.093 | 0.189 | 0.078 |
| | Vicuna-16k | 2 Shot | 0.063 | 0.440 | 0.035 | 0.090 | 0.124 | 0.094 |
| | | 5 Shot | 0.073 | 0.360 | 0.042 | 0.093 | 0.143 | 0.089 |
| 20B | InternLM | 2 Shot | 0.419 | 0.496 | 0.392 | 0.135 | 0.151 | 0.136 |
| | | 5 Shot | 0.453 | 0.491 | 0.455 | 0.132 | 0.154 | 0.127 |
| 33B | Vicuna | 2 Shot | 0.411 | 0.487 | 0.406 | 0.159 | 0.191 | 0.155 |
| | | 5 Shot | / | / | / | / | / | / |
| 34B | CodeLlama | 2 Shot | 0.297 | 0.487 | 0.218 | 0.078 | 0.150 | 0.096 |
| | | 5 Shot | 0.243 | 0.438 | 0.178 | 0.073 | 0.140 | 0.119 |
| 40B | Falcon | 2 Shot | 0.305 | 0.465 | 0.251 | 0.155 | 0.183 | 0.154 |
| | | 5 Shot | / | / | / | / | / | / |
| 70B | Llama-2 | 2 Shot | 0.528 | 0.468 | 0.708 | 0.126 | 0.175 | 0.189 |
| | | 5 Shot | 0.581 | 0.472 | 0.837 | 0.118 | 0.209 | 0.237 |
| | Platypus2 | 2 Shot | 0.166 | 0.562 | 0.102 | 0.119 | 0.138 | 0.131 |
| | | 5 Shot | 0.212 | 0.432 | 0.149 | 0.135 | 0.194 | 0.130 |
| / | GPT-3.5 | 2 Shot | 0.128 | 0.490 | 0.076 | 0.088 | 0.111 | 0.094 |
| | | 5 Shot | 0.131 | 0.437 | 0.079 | 0.103 | 0.113 | 0.123 |
| | GPT-4 | 2 Shot | 0.187 | 0.501 | 0.120 | 0.159 | 0.187 | 0.207 |
| | | 5 Shot | 0.175 | 0.509 | 0.111 | 0.136 | 0.157 | 0.141 |

# J  ABLATION STUDY ON PROVIDED INFORMATION

We conduct some ablation studies on how different information provided to LLMs is affecting their performance. We compare the impact of manually reversed decompiled code in the CTF dataset, the impact of only decompiled code provided in the MAGMA dataset, and the impact of larger context provided instead of only a single function provided in both the CTF and MAGMA datasets. The result of the CTF dataset is shown in Figure 8 and the result of the MAGMA dataset is shown in Figure 9.

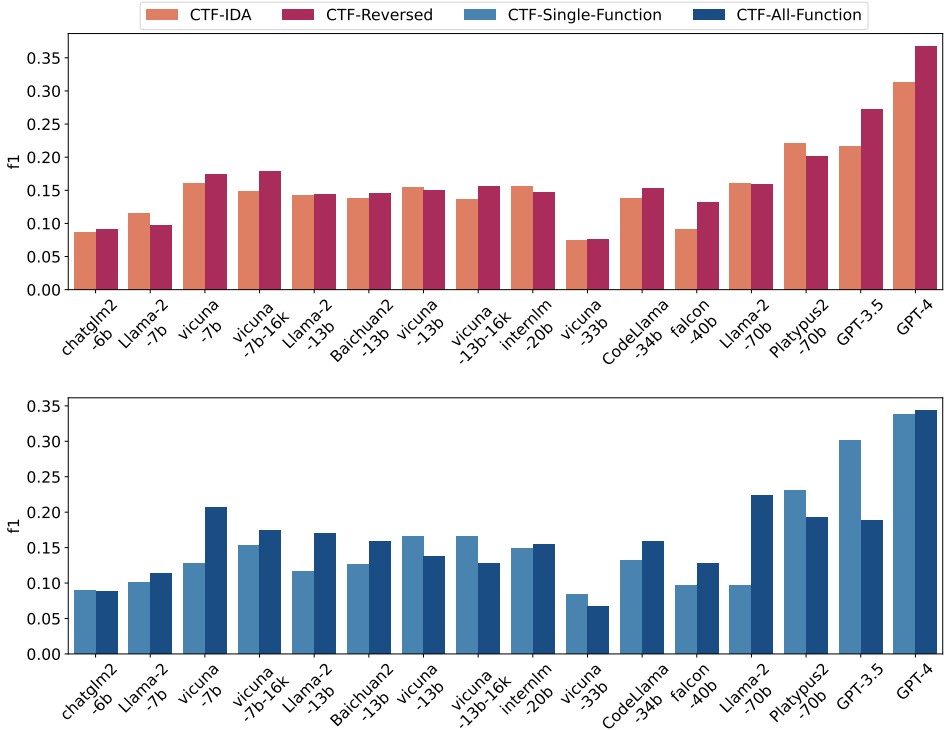

Figure 8: CTF's F1 score averaged over 2 types of shots (2 shots and 5 shots) and 2 types of classification (Binary classification and multi-class classification). Upper: Comparison between raw decompiled code from IDA (`CTF-IDA`) and manually reversed decompiled code (`CTF-Reversed`). Lower: Comparison between only a single function is provided to the LLM (`CTF-Single-Function`) and all functions in a binary are provided to the LLM (`CTF-All-Function`).

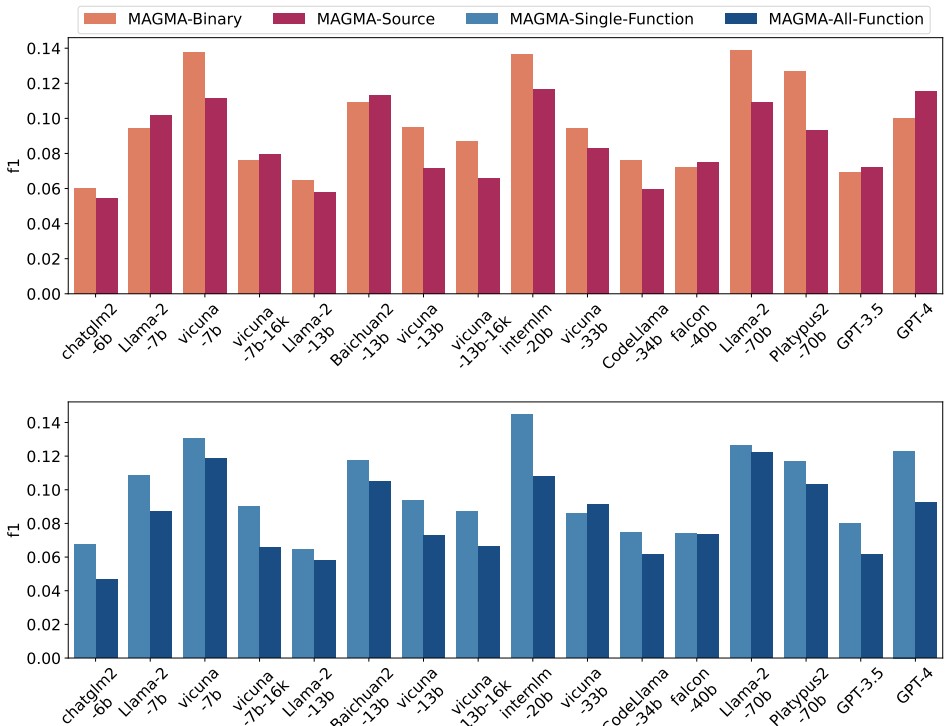

Figure 9: MAGMA's F1 score averaged over 2 types of shots (2 shots and 5 shots). Upper: Comparison between raw decompiled code (`MAGMA-Binary`) and source code (`MAGMA-Source`). Lower: Comparison between only a single function is provided to the LLM (`MAGMA-Single-Function`) and all functions related to the vulnerability are provided to the LLM (`MAGMA-All-Function`).

## K  BAD CASES OF DECOMPILED CODE

In this section, we show two common examples where IDA fails to recover the semantics of the original source code by decompiled code. As shown in Figure 10, line `v0 = alloca(...);` in decompiled code actually corresponds to `sub rbp, rax` in assembly code, which means that the decompiled code translates the assembly code incorrectly, ignoring that it is a dynamic stack allocation. And in Figure 11, IDA 7.6 cannot understand the `switch` statement generated by a newer compiler, resulting completely wrong decompiled code (A bare `jmp rax`). This requires extra manuall effect to fix in a newer IDA.

Decompile code

Assembly code

Figure 10:  Decompiled code and assembly code of function `calc_sum` in CTF challegnge `zer0pts_2020_protrude`.

Decompile code

Assembly code

Figure 11: Decompiled code and assembly code of function containing `switch` in IDA Pro 7.6 and 8.3.

