# OpenReview forum: "How Far Have We Gone in Vulnerability Detection Using Large Language Model"
_ICLR.cc/2024/Conference — Submitted to ICLR 2024_

### Official Review · Reviewer_91Jf · 2023-10-19

**Soundness:** 3 good
**Presentation:** 3 good
**Contribution:** 2 fair
**Rating:** 5
**Confidence:** 4

**Summary:**

Vulnerability detection's primary goal is to discover software security threats, which is essential for mitigating cyber-attacks. The authors present a study on the efficacy of Large Language Models (LLMs) in vulnerability detection. By selecting a variety of LLMs, including GPT-3.5 and GPT-4, as well as other open source models, the authors compare the performance of these LLMs against deep learning models and static analysis tools. The benchmarks consist of various datasets, both artificial like CTF and real-world datasets.

**Strengths:**

The authors have chosen a diverse range of models, encompassing popular GPT versions, open-source models. This wide variety ensures a thorough comparison for LLMs.

The inclusion of real-world datasets ensures practical relevance. By comparing performance on artificial datasets like CTF versus real-world datasets, the paper provides a holistic view of LLM capabilities.

It's commendable that the authors also address the limitations of LLMs, especially in real-world scenarios where context might be lacking or in decompiled code scenarios.

**Weaknesses:**

While the paper does compare performances between different LLMs, the "why" behind these performances could be elaborated upon. Understanding the intricacies of each model might explain why some models performed better than others. For example, architectural nuances, the type and quality of training data, or the model's inherent design could influence performance.

Also the baseline models (traditional deep learning and static analysis tools) could be explored in more depth. More insights into why and where they outperformed or underperformed compared to LLMs would be valuable.

When LLMs incorrectly classify vulnerabilities, understanding the nature of these mistakes (whether they are false positives or false negatives) would be invaluable. This could be complemented by representative examples to highlight common pitfalls the models encounter.

The paper does discuss the limitations of decompiled code, but a deeper dive into how these limitations impacted the results and the potential solutions or workarounds would add value.

Minor Issues:
The reference should have been revised. Some preprints have been published, e.g., 'CodeXGLUE: A Machine Learning Benchmark Dataset for Code Understanding and Generation' is published on NeurIPS 2021.

Abbreviation should be consistent e.g., 'Cve' and 'CVE'.

Typo in reference '$\mu$'

**Questions:**

I'm just curious if LLMs can perform well on zero-day vulnerabilities.

Could you provide some examples of false positives or false negatives?

---

> ### Author Response · Authors · 2023-11-21
> **Response to Reviewer 91Jf**
>
> ## Response to Weaknesses
>
> 1. In Appendix Section H (Analysis of the Models' Performance), we illustrated the models' capability to assess different types of vulnerabilities across various datasets using confusion matrix. Given our different levels of knowledge about the training data for various pretrained models, we may not be able to directly compare differences arising from training data. However, a significant difference we noticed is the bias from alignment methods used, where models trained with RLHF tend to favor certain outputs. At the same time, baseline models often perform better than LLMs on real-world datasets, likely because they attempt to provide some results from being trained on vulnerability datasets, whereas LLMs often give a very conservative result, "No Vulnerability," leading to poorer performance that might require further finetuning. We attempt to clarify this phenomenon in Section 4.4 (Real-World Dataset).
> 2. We've added a challenging real-world example to Section F.4, where understanding the boundaries of different data types clearly is required, as well as identifying where the problem occurs—this is relatively difficult even for humans and may necessitate additional program data flow slicing to help LLMs focus on the potential vulnerability path before making a judgment.
> 3. An additional example has been added to Section K, highlighting the limitations of decompiled code. However, it clarifies that part of these limitations stem from the decompiler's inadequate support for patterns from newer compilers, and there is a need for decompilers to handle a wider array of scenarios. Moreover, in Section 5.2 (Limitation of Decompiled Code), we discussed possibilities of using LLMs with well-encoded features directly, similar to LLaVA, for subsequent vulnerability detection.
> 4. We appreciate the reviewer highlighting this issue, and we have tried our best to address in the revision.
>
> ## Response to Questions
>
> 1. Based on the results from real-world datasets and the confusion matrix shown in Section H (Analysis of the Models' Performance), LLMs lead to a high number of false positives and false negatives, suggesting that they may not yet be wholly reliable for automated 0day vulnerability dection and might require additional verification tools, such as fuzzing and static analysis, or introducing humans in the loop to assist with vulnerability detection. This method is commonly used now, and the new paradigm for fully automated vulnerability detection by LLMs is still to be explored.
> 2. In the revision's Section F.4, we included examples of LLM's mistakes to show the complexity involved in vulnerability detection within real-world programs.

---

> > ### Comment · Reviewer_91Jf · 2023-11-22
> >
> > Thank you for the response. I am keeping my score unchanged for now.

---

### Official Review · Reviewer_Ni1D · 2023-10-31

**Soundness:** 2 fair
**Presentation:** 2 fair
**Contribution:** 2 fair
**Rating:** 5
**Confidence:** 4

**Summary:**

The paper introduces a vulnerability benchmark for investigating the capabilities of Large Language Models (LLMs) in vulnerability detection.

The authors conduct extensive experiments involving existing solutions, assessing 16 LLMs and 6 state-of-the-art (SOTA) methods in vulnerability detection. Via the authors’ claim, the evaluation result uncovers a paradox in performance levels and highlights the untapped potential of LLMs.

**Strengths:**

The authors have introduced a combined dataset for evaluating LLMs’ vulnerability detection abilities. They have designed and conducted a comprehensive evaluation process to assess the vulnerability detection capabilities of Language Models (LLMs).

**Weaknesses:**

The claim “We thoroughly analyze their strengths and weaknesses in vulnerability detection tasks, identifying areas for improvement and future research directions” is not clearly explained case by case in the paper. The authors mainly focus on ChatGPT. How about other LLMs used in the paper? Notably, what are the areas for improvement and future research directions?

The finding relevant to “the lack of context” in the statement “on larger software platforms, due to the lack of context, LLMs do not sufficiently comprehend vulnerabilities” is one of the well-known limitations of LLMs in text data. LLMs strongly rely on the relevant context instead of the data themselves. The lack of context of the appearance of irrelevant context strongly negatively affects the LLM's performance.

The novelty of the proposed framework (not applicable in the paper because the paper was not going to propose any innovative framework for vulnerability detection) or dataset is limited. The introduced dataset is simply from a combination of some datasets.

The aspect that is relevant to the characteristics in terms of the semantic and syntactic relationships of the source code data is not mentioned or studied. From many state-of-the-art deep learning-based vulnerability detection methods, to deal with vulnerability detection, the models need to be successful in leveraging the semantic and syntactic relationships between the code tokens and source code statements. That helps the model figure out potential vulnerabilities in the data to distinguish the vulnerable and benign data. Failing to learn the important properties of the source code data can also be another limitation of LLMs in vulnerability detection.

**Questions:**

There are many big and well-known datasets (consisting of various types of vulnerability) ready to be used for vulnerability detection, such as Big-Vul (Fan et al., 2020b) and DiverseVul (Chen et al., 2023). What are the advantages of the introduced dataset compared to these ones?

What are the main strengths and weaknesses of LLMs in vulnerability detection case by case and in general found in the paper? What are your corresponding suggestions to deal with these limitations?

The combination of some datasets to form the used dataset, VulBench, are random or are there any insightful intuitions for that?

---

> ### Author Response · Authors · 2023-11-21
> **Response to Reviewer Ni1D**
>
> ## Response to Weakness:
>
> 1. We thank the reviewer for pointing out this oversight, and we have supplemented our paper's Evaluation section with additional discussions about other models. In Section H (Analysis of the Models' Performance), we delve deeper into the behavior of different models and have identified biases and an over-conservatism that are prevalent among many open access large models. Furthermore, we compared the models' ability to process large contexts by providing them with differing amounts of information.
> 2. While the lack of context is a widely recognized limitation of current LLMs, our design of the vulnerability dataset takes into consideration the LLM's capabilities when it is not lacking context related to vulnerabilities. This includes providing all functions within a binary in CTF, enabling LLMs to analyze the entire binary, and in MAGMA, offering additional context required for human analysis as input to the models. These two aspects assess the capabilities of models when they are not lacking context relevant to vulnerabilities, which is inconsistent with past vulnerability datasets. In real-world datasets, we also observed that despite providing relevant context, the presence of irrelevant context has a strongly negative effect on LLM's performance, as stated by the reviewer.
> 3. We pointed out in **Clarification on Dataset Distinction.**
> 4. This is an intriguing question, but in the LLM context, we often provide plain text to the models as input, without the ability to include semantic and syntactic relationships explicitly. Yet, LLMs are capable of generating grammatically correct and semantically appropriate code in many downstream tasks, which indicates that they do not necessarily require explicit semantic and syntactic inputs to understand the code properly. Recent works based purely on transformers, like VulBERTa (Hanif & Maffeis, 2022) and LineVul (Fu & Tantithamthavorn, 2022), have shown that vulnerability detection can be effectively performed without explicitly providing semantic and syntactic relationships, and still outperform tools like VulDeeLocator and VulDeePecker, which do provide explicit relationships between code tokens and source code statements.
>
> ## Response to Questions:
>
> 1. LLMs often have difficulty attending well when facing large functions or excessive context, a challenge that static analysis tools can mitigate by analyzing slices of a program or conducting localized analysis based on the program's data flow graph, thus avoiding focus drift. This issue could be addressed through a combination of static analysis and LLM vulnerability detection, providing only part of the program to help maintain the model's focus. Moreover, the limited processing capability of context leads to a conservative model behavior in real-world programs, continuously outputting "No Vulnerability."
> However, LLMs can serve as knowledge base due to their extensive pretraining, aiding the vulnerability detection process. For example, in Section F.3, the LLM identifies a potential Use After Free issue with the input `context_->CallFunction`, even without context, by learning from its usage.
> 2. We include the CTF dataset, which offers a relatively complete context to simulate vulnerabilities that occur in actual software. MAGMA, a dataset derived from fuzzing, includes all conditions that trigger and fix vulnerabilities, which helps in understanding and providing proper labeling. Also, all vulnerable functions in MAGMA dataset are contained within the same binary, making decompiled code extraction feasible, simulating closed-source software vulnerability detection. This differs from datasets such as D2A, Devign, and Big-Vul, which originate from different commits within a project and are relatively harder to compile to obtain corresponding decompiled code. D2A, Devign, and Big-Vul are also commonly used in past works, which is why we incorporated them into our dataset. Through this combination, we can provide a variety of context sizes and simulate targets like closed-source software vulnerability detection.

---

### Official Review · Reviewer_YnvL · 2023-10-31

**Soundness:** 2 fair
**Presentation:** 2 fair
**Contribution:** 2 fair
**Rating:** 3
**Confidence:** 4

**Summary:**

This paper presents a benchmark called VulBench for evaluating the performance of Large Language Models (LLMs) and state-of-the-art methods in automated vulnerability detection. The benchmark comprises vulnerability data collected from Capture The Flag (CTF) challenges, security flaws reported by fuzzing tools, and existing vulnerability detection benchmarks, with annotations specifying the type and root cause of each vulnerability. The paper conducts extensive experiments involving 16 LLMs and 6 state-of-the-art methods to assess their effectiveness in detecting vulnerabilities. The results reveal a paradox in performance levels and suggest that LLMs have untapped potential in this domain.

**Strengths:**

+ This paper sheds light on new resources to collect software vulnerabilities to evaluate the automated vulnerability detectors. They identify two useful resources, CTF and fuzzing reported security flaws, which can potentially compensate for the diversity of the common strategy to focus on Github commits to extract vulnerable and benign code snippets. Also, the well-controlled CTF challenges have few label noises compared to samples collected from Github commits, since the changed function in a commit might not directly relate to the vulnerability while they are typically sampled and noisily labeled by existing benchmarks.
+ This paper conducts extensive evaluation on 16 models with up to tens/hundreds of billions of parameters while existing works mostly evaluate smaller code LMs with at max hundreds of millions of parameters. The experiments tend to reveal a more up-to-date SOTA performance from the most capable LLMs, though even these latest models seem not to significantly outperform the smaller models and not promising enough in vulnerability detection.

**Weaknesses:**

__The main contributions of VulBench are neither clearly specified nor sufficiently evaluated.__ As a datasets/benchmark paper, the most important contribution should be the additional value it brings, compared to the existing benchmarks of the same type. However, such contributions are not clear in this paper.

First, though the paper identifies new resources to collect vulnerable samples, it is not clear how different and valuable these new resources and samples are, compared to the existing benchmarks. I would recommend the author illustrate the value of these samples, such as whether they cover unique CWEs that existing benchmarks do not have, or whether they compensate for specific types of low-resource vulnerabilities, etc. In addition, in Section 3.2.3, it is quite vague how VulBench cleans up Devign, D2A, and BigVul, and also not clear how accurate the labels are after their filtering. I would recommend the authors to concretize the effects of the cleaning and filtering, such as what was the ratio of noisy labels in the original benchmark and how does that improve with VulBench's version.

Second, the comparison between VulBench and existing benchmarks is missing. To illustrate the value of the new benchmark, the most effective way is to directly compare with the existing benchmarks to explain what are the difference. However, the evaluation of this paper focuses on comparing the CTF split and real-world split of its own benchmark and ignores to compare VulBench, as a whole, to Devign, D2A, and BigVul. I would suggest the authors to evaluate the 16 LLMs on these existing benchmarks as well and conduct a thorough analysis to reveal what perspectives could not be well studied by existing benchmarks, while VulBench's additional resources and sample filtering help, serving as a more comprehensive and accurate evaluation of LLMs' capacity in vulnerability detection than others.

__The dataset contains many reversed decompiled code, questioning the naturalness and reality of these code samples.__ While I agree that the samples from CTF and MAGMA bring better diversity than focusing on Github commit, the decompiled code samples from these resources are concerning. The decompiled source code could be quite different from realistic programs written by human, and there could be instinct patterns or data structures that are hardcoded by the decompilation tool but rarely or never used by the developers. Though the authors mentioned that they try their best to make the decompiled samples look natural by variable renaming, etc, it is not clear how effective their decorations are to bring back the code naturalness. I would encourage the authors to quantify, beyond only case studies, how (un)natural these decompiled samples are compared to human-written code, and this is important to estimate the usefulness and reality of the benchmark to evaluate LLM's capability in vulnerability detection in the real-world scenario.


__The main methodology of this paper, the dataset construction process, is rather brief and vague, missing details and illustrations for understanding.__ In general, Section 3.2, as the explanation of the main methodology is not understandable and lacks details. For 3.2.1 and 3.2.2, it is better to assume that audiences have no background in CTF problems and fuzzing, so more details should be explained (maybe in Appendix), such as what are the format of CTF problems and fuzzing reports, and how the labels are constructed accordingly. A few concrete examples from the raw data to the benchmark samples will be appreciated. This will not only increase the readability but also the reliability of the sample labels.

__The benchmark is not available for review so far.__ Somehow I could not find the link to this benchmark. For this paper, the benchmark itself is the major output, and I might need to manually inspect the quality of tens of samples to determine the general quality of this work. Due to the brief and vague description of the approach, I would urge the authors to anonymously release the benchmark for reviewers to directly evaluate the quality. Of course, if I accidentally missed the link somewhere, please correct me.

**Questions:**

- Can the authors provide more details of how CTF dataset is formulated, like what the raw challenge looks like, and what information will be exacted for labeling, etc?

- What is the ratio of noisy labels being removed by VulBench from Devign, D2A, and Big-Vul?

- Will the authors anonymously release the benchmark for review?

---

> ### Author Response · Authors · 2023-11-21
> **Response to Reviewer YnvL**
>
> ## Response to Weaknesses
>
> 1. We discuss the main contributions of VulBench in the **Clarification on Dataset Distinction** of **General Response to All Reviewers**.
> 2. In both the CTF dataset and the MAGMA dataset, we provide decompiled code as one of the inputs to the models. However, for the CTF dataset, the challenges typically do not come with source code, so we can only obtain the decompiled code corresponding to the challenges and attempt to approximate the original source code through manual reverse engineering. In real-world scenarios, vulnerability detection often lacks access to source code, necessitating reverse engineering. This is common in situations for Windows softwares and macOS softwares. Likewise, since we do not have access to the original source code of the CTF challenges, we cannot determine how far our decompiled code deviates from the real source code. We do manually reverse engineering as to explore the ability for vulnerability discovery with LLM. Yet, as shown in Figure 8 of the revised paper, providing this manually reversed code seems to improve the ability of different models in vulnerability detection. For the MAGMA dataset, where we have the actual source code, we did not perform any manual reverse engineering but used both the source code and the decompiled code without manual reverse as inputs for the models.
> However, as the reviewer stated, decompiled code is quite different from realistic programs written by human developers and it contains instinct patterns or data structures that are hardcoded by the decompiler but rarely or never used by developers. We clarified this issue in Section 3.2.1 regarding the construction of the CTF dataset.
> 3. We further elaborated on the construction process for each dataset in Section 3 Dataset and provided examples from different datasets in Appendix Section E Example in Each Dataset for better readability by readers without a background in CTF or Fuzzing.
> 4. Thank you to the reviewer for pointing this out. We have provided our dataset for review.
>
> ## Response to Question
>
> 1. Each CTF challenge consists solely of an executable binary program (e.g Notepad.exe, Taskmgr.exe). We extracted their decompiled code using IDA Pro and then performed subsequent processing, such as manual reverse engineering. We provided examples of decompiled code obtained from raw challenges in Section E.1 CTF Dataset. You can also check the datasets we provide for review.
> 2. We refer to past results from measuring other vulnerability datasets (Croft et al. (2023)), with accuracies of 0.8 (Devign), 0.543 (Big-Vul), and 0.286 (D2A), and through manual annotation, we aim to eliminate as much of the erroneous data noise as possible.
> 3. We have uploaded the dataset to [anonymous github](https://anonymous.4open.science/r/VulBench-EA6F/) for review

---

> > ### Comment · Reviewer_YnvL · 2023-11-22
> >
> > Thank you for your response. While I appreciate the revision to the paper, my general attitude towards this paper still remains. I am keeping my score unchanged for now.

---

### Official Review · Reviewer_FRwA · 2023-11-01

**Soundness:** 3 good
**Presentation:** 2 fair
**Contribution:** 2 fair
**Rating:** 5
**Confidence:** 4

**Summary:**

A new vulnerability dataset derived from CTF challenges and real-world applications is proposed in this work. The dataset provides annotations of each vulnerable function with the vulnerability type and descriptions of root cause of the vulnerability with a goal to enable improved evaluation of LLMs’ capabilities in vulnerability detections. Additionally, this paper evaluates 16 LLMs and 6 SOTA models using the proposed benchmark and presents some insights about LLMs performance levels with few shot prompts and increased context windows.

**Strengths:**

* Investigation of LLMs capabilities and shortcomings with respect to vulnerable code identification is a critical challenge with great potential for future innovations.

* The proposed benchmark includes both synthetic and real world vulnerabilities. Isolation of synthetic and real world vulnerabilities in performance analysis provides useful insights.

**Weaknesses:**

The proposed work combines different synthetic and exisiting real world vulnerabilities and adds annotations to evaluate LLMs. However, it is not clear how comprehensive is the new dataset. I think the work lacks evidences on two aspects of the dataset:
1) Vulnerability coverage: how much coverage the proposed dataset have in different types of vulnerabilities?
2) Advantages over existing benchmarks: Extensive experiments are performed on LLMs and compartive analysis with other DL and static analysis methods are presented. However, it is not clear what new and critical insights one can derive using the proposed benchmark compared to existing dataset like MT-bench (Zheng et al., 2023a) or dataset used in Cheshkov et al. (2023).

**Questions:**

1. Is there any new vulnerability added that was not part of any of the previous benchmarks?
2. It would be interesting to see if there is any specific vulnerability class where LLMs have increased detection capabilities. Do you have any insights based on the experiments conducted in this work?
3. Could you elaborate the inputs used in Figure-5? More specifically, what does “providing all functions” indicate? How are the context limitations maintained in this setup?
4. Nit: I think ‘multi-classification’ is not a standard terminology. A more standard and specific term like “multi-label” or “multi-class” classification would provide increased clarity of the evaluation approach.

---

> ### Author Response · Authors · 2023-11-21
> **Response to Reviewer FRwA**
>
> ## Response to Weakness:
>
> 1. In Section C (Dataset Details), we have added the types of vulnerabilities included in our dataset.
> 2. In contrast to MT-Bench used in Cheshkov et al. (2023), our dataset pays closer attention to memory vulnerabilities and evaluates many different open access models. We discuss the insights in the aboding **Key Insights from Our Experiments** in **General Response to All Reviewers**.
>
> ## Response to Questions:
>
> 1. Our study did not introduce new types of vulnerabilities, but we put a greater efforts on the quality of the dataset.
> 2. In the paper's Appendix Section H (Analysis of the Models' Performance), through confusion matrices, we illustrated the models' ability to assess different types of vulnerabilities on various datasets. This allows us to see more intuitively where models are prone to errors with certain types of vulnerabilities and where they perform better. On datasets like CTF, models handle Buffer Overflow and Format String vulnerabilities better, and on real-world datasets, they demonstrate better performance for vulnerabilities like Buffer Overflow that appear more frequently in training corpora, yet they are still far from an acceptable level of usability.
> 3. The term 'all functions' refers to our providing all functions within an executable binary program to the model, as the full_context.c we present in the ctf datasets (e.g https://anonymous.4open.science/r/VulBench-EA6F/data/ctf/blag_tjctf_2016/full_context.c) . We have updated the caption in Figure 8 in the revision. As for the context limitations, in the CTF dataset, the number of functions in a binary is relatively small, allowing the model to analyze all available functions directly. It is for this reason that we rarely encountered situations where it was not feasible to provide all functions to the model.
> 4. We greatly appreciate the identification of this issue, and we have fixed it in the revision.

---

> > ### Comment · Reviewer_FRwA · 2023-11-22
> > **Thanks for your detailed response.**
> >
> > Thank you for answering my queries. The details of on how the new benchmark is different from existing ones are clarified. However, I think evaluations can be improved to demonstrate any increased effectiveness over existing benchmarks.

---

> > > ### Author Response · Authors · 2023-11-22
> > > **Question about the 'demonstrate any increased effectiveness'**
> > >
> > > Thanks for your response. Could you please specify if you are suggesting include an assessment in Evaluation on the accuracy of the human-annotated labels within our dataset? We seek to understand if this is what you refer to under 'increased effectiveness.' In case there is a different aspect you are alluding to, we would be thankful for any further details you might provide.

---

### Author Response · Authors · 2023-11-21
**General Response to All Reviewers**

We sincerely thank you for your insightful reviews and valuable feedback on our paper. We appreciate the opportunity to address the concerns and questions raised.

**Clarification on Dataset Distinction**

Firstly, we would like to clarify the distinction of our dataset from previous datasets in the field. While past research has utilized automated mechanisms to gather large datasets for vulnerability detection, a high-quality benchmark specifically tailored for this purpose has been lacking. In our paper, we discuss the findings from Croft et al. (2023), who noted the insufficient accuracy levels of datasets such as Devign, Big-Vul, and D2A. Specifically, their research indicated accuracy rates of 0.8 for Devign, 0.543 for Big-Vul, and only 0.286 for D2A, underscoring the lack of precision in these datasets. Similarly, in DiverseVul (Chen et al., 2023), manually evaluated accuracies were 0.6 for DiverseVul, 0.25 for BigVul, and 0.478 for CrossVul. **These datasets, therefore, are inadequate as benchmarks to evaluate the effectiveness of vulnerability detection tools**, including LLMs, other deep learning baselines, and static analysis tools. The datasets, often automatically generated, are prone to include false positives, such as non-functional changes in a CVE patch commit, like code formatting. If the original dataset labels are incorrect, the results derived from them are not convincing.

To accurately evaluate the performance of LLMs and existing SOTA approaches in vulnerability detection—a task that requires extensive expert knowledge—we addressed these issues through a manual annotation approach. Our dataset was labeled by experienced security analysis experts and world-class CTF team members. Our dataset differs from previous work in the following ways:

1. Ensuring quality for evaluating the performance of vulnerability detection tools.
2. Providing natural language descriptions of vulnerability root causes. While we lack the capacity to evaluate these model outputs on a large scale, we lay the groundwork for future work.
3. Differing from previous datasets that contained only single functions, our dataset provides more context related to vulnerabilities, including complete binary functions in CTF and function calls, macro definitions, and structure information in MAGMA related to the vulnerabilities.

**Dataset for Review**

The dataset is available for anonymous review at: [VulBench Dataset](https://anonymous.4open.science/r/VulBench-EA6F/README.md)

**Key Insights from Our Experiments**

1. In the context of vulnerability detection, we found that LLMs have a distinct advantage over the current SOTA methods, including those based on deep learning and program analysis. LLMs, despite not being specifically trained for vulnerability detection scenarios, performed better or comparably to these SOTA methods. Additionally, the use of LLMs in vulnerability detection demonstrated strong interpretability. This once again proves the robust generalization capability of LLMs.
2. Current LLMs exhibit significantly higher precision in identifying common vulnerabilities (those more likely to appear in pre-training corpora) compared to other types. This indicates that with the provision of higher quality and more diverse vulnerability datasets for LLM training, there is potential for LLMs to perform even better in this field.
3. A very interesting insight, as analyzed in Section H, is that LLMs trained with different methods exhibited highly different behaviors in real-world vulnerability discovery scenarios. Supervised fine tuning (SFT) models tend to output 'No Vulnerability', such as Platypus and Vicuna. Some RLHF models (such as Llama-13B, Llama-70B, Internlm 20B, and Baichuan2) tend to output widespread vulnerabilities (Use-after-free or Buffer-overflow). However, GPT-4 and GPT-3.5, also trained with RLHF, still tend to output 'No Vulnerability' if they can’t identify the vulnerability. We hypothesize that the discrepancy arises because SFT aligns the models to output definitive content, and if no vulnerability is detected, they output 'No Vulnerability'. The RL training objective encourages models to output something for more rewards, often leading to the output of something rather than nothing, thereby making the model more prone to illusion and randomly output a prevalent vulnerability. The less pronounced hallucinations issue in GPT-4 and GPT-3.5 may be due to OpenAI's training methods for these models, which possibly included measures to avoid hallucinations and prevent the model from collapsing into certain biases. **In the future, when using LLMs for vulnerability detection, eliminating this kind of bias will be crucial, as it can significantly improve precision.**

---

### Author Response · Authors · 2023-11-21
**Updates in the Revised Paper**

We have also made significant revisions to the paper, as outlined below:

- **Section 1 (Introduction):** Updated to highlight the limitations of past vulnerability datasets and the contributions of our dataset.
- **Section 3.2 (Dataset Construction):** Detailed description of dataset construction, with updates in Section C (Dataset Details) on types of vulnerabilities included, and in Section E (Examples in Each Dataset), where we provide examples from each dataset and compare raw decompiled code with manually reversed decompiled code, along with examples of natural language descriptions of vulnerabilities.
- **Section 4 (Evaluation) and Section H:** Added more analyses of open-access models, not just focusing on OpenAI's GPT models. Section 4.4 (Analysis of the Models' Performance) now includes an analysis of model performance across multiple datasets in multi-classification tasks, with full confusion matrix results presented in Section H.
- **Section J (Ablation Study on Provided Information):** Updated captions and provided more explanations on single function versus all functions.
- **Section K (Bad Cases of Decompiled Code):** Added examples where decompiled code has limitations, noting that these issues have been fixed in newer versions of IDA, demonstrating the evolving capability of LLMs with advancements in decompilers.
- **Section 5.2 (Limitation of Decompiled Code):** Discussed potential solutions to limitations of decompiled code, including the use of an assembly encoder for the models.
- **Section 6 (Conclusion):** Updated the conclusion based on the experimental results presented in the Evaluation section.

---

### Meta-Review · Area_Chair_htz6 · 2023-12-06

**Metareview:**

The paper presents VulBench a benchmark for automated vulnerability detection. The benchmark aggregates vulnerability data collected from multiple sources. The paper conducts extensive experiments involving 16 LLMs and 6 state-of-the-art methods to assess their effectiveness in detecting vulnerabilities.

## Strengths

- Looks into a useful problem
- An extensive evaluation over many LLMs.

## Weaknesses
- A vulnerability detection benchmark should comprehensively tackle the false positive issue that most existing methods suffer from. This isn’t significantly considered in this version of the work.
- It seems unlikely that a large portion of the benchmark has not leaked into the training set of the evaluated LLMs. This work doesn’t take any measures against that.
- The added value from the aggregation of the datasets isn’t clear. The paper would need to better persuade the community that the benchmark has the desirable properties needed in order to become established.

Although I am generally predisposed in releasing benchmarks and datasets, I believe that additional work is required before accepting this work. Therefore, I suggest rejecting the paper at its current state.

**Justification For Why Not Higher Score:**

While the presented benchmark could be useful, at its current state it won't probably allow to accurately gauge progress in the area of vulnerability detection with LLMs.

**Justification For Why Not Lower Score:**

n/a

---

### Decision · Program_Chairs · 2024-01-16

Reject